# Jointly-Trained State-Action Embedding for Efficient Reinforcement Learning

## Abstract

While reinforcement learning has achieved considerable successes in recent years, state-of-the-art models are often still limited by the size of state and action spaces. Model-free reinforcement learning approaches use some form of state representations and the latest work has explored embedding techniques for actions, both with the aim of achieving better generalization and applicability. However, these approaches consider only states or actions, ignoring the interaction between them when generating embedded representations. In this work, we propose a new approach for jointly learning embeddings for states and actions that combines aspects of model-free and model-based reinforcement learning, which can be applied in both discrete and continuous domains. Specifically, we use a model of the environment to obtain embeddings for states and actions and present a generic architecture that uses these to learn a policy. In this way, the embedded representations obtained via our approach enable better generalization over both states and actions by capturing similarities in the embedding spaces. Evaluations of our approach on several gaming, robotic control, and recommender systems show it significantly outperforms state-of-the-art models in both discrete/continuous domains with large state/action spaces, thus confirming its efficacy and the overall superior performance.

## 1 Introduction

*Reinforcement learning* (RL) has been successfully applied to a range of tasks, including challenging gaming scenarios (Mnih et al., 2015). However, the application of RL in many real-world domains is often hindered by the large number of possible states and actions these settings present. For instance, resource management in computing clusters (Mao et al., 2016; Evans & Gao, 2016), portfolio management (Jiang et al., 2017), and recommender systems (Lei & Li, 2019; Liu et al., 2018) all suffer from extremely large state/action spaces, thus challenging to be tackled by RL.

In this work, we investigate efficient training of reinforcement learning agents in the presence of large state-action spaces, aiming to improve the applicability of RL to real-world domains. Previous work attempting to address this challenge has explored the idea of learning representations (embeddings) for states or actions. Specifically, for state embeddings, using machine learning to obtain meaningful features from raw state representations is a common practice in RL, e.g. through the use of convolutional neural networks for image input (Mnih et al., 2013). Previous works such as by Ha & Schmidhuber (2018b) have explored the use of environment models, termed *world models*, to learn abstract state representations, and several pieces of literature explore state aggregation using bisimulation metrics (Castro, 2020). While for action embeddings, the most recent works by Tennenholtz & Mannor (2019) and Chandak et al. (2019) propose methods for learning embeddings for discrete actions that can be directly used by an RL agent and improve generalization over actions. However, these works consider the state representation and action representation as isolated tasks, which ignore the underlying relationships between them. In this regard, we take a different approach and propose to jointly learn embeddings for states and actions, aiming for better generalization over both states and actions in their respective embedding spaces.

To this end, we propose an architecture consisting of two models: a model of the environment that is used to generate state and action representations and a model-free RL agent that learns a policy using the embedded states and actions. By using these two models, our approach combines aspects

of model-based and model-free reinforcement learning and effectively bridges the gap between both approaches. In contrast to model-based RL, however, we do not use the environment model for planning, but to learn state and action embeddings. One key benefit of this approach is that state and action representations can be learned in a supervised manner, which greatly improves sampling efficiency and potentially enables their use for transfer learning. In sum, our key contributions are:

- We formulate an embedding model for states and actions, along with an internal policy $\pi_i$ that leverages the learned state/action embeddings, as well as the corresponding overall policy $\pi_o$. We show the existence of an overall policy $\pi_o$ that achieves optimality in the original problem domain.

- We further prove the equivalence between updates to the internal policy $\pi_i$ acting in embedding space and updates to the overall policy $\pi_o$.

- We present a supervised learning algorithm for the proposed embedding model that can be combined with any policy gradient based RL algorithms.

- We evaluate our methodology on some game-based as well as real-world tasks and find that it outperforms state-of-the-art models in both discrete/continuous domains.

The remainder of this paper is structured as follows: In Section 2, we provide some background on RL. We then give an overview of related work in Section 3, before presenting our proposed methodology in Section 4, which we evaluate in Section 5.

## 2 BACKGROUND

We consider an agent interacting with its environment over discrete time steps, where the environment is modelled as a discrete-time *Markov decision process* (MDP), defined by a tuple $(\mathcal{S}, \mathcal{A}, \mathcal{T}, \mathcal{R}, \gamma)$. Specifically, $\mathcal{S}$ and $\mathcal{A}$ are the sets of all possible states and actions, referred to as the *state space* and *action space*, respectively. In this work, we consider both discrete and continuous state and action spaces. The transition function from one state to another, given an action, is $\mathcal{T} : \mathcal{S} \times \mathcal{A} \mapsto \mathcal{S}$, which may be *deterministic* or *stochastic*. The agent receives a reward at each time step defined by $\mathcal{R} : \mathcal{S} \times \mathcal{A} \mapsto \mathbb{R}$. $\gamma \in [0, 1]$ denotes the reward discounting parameter. The state, action, and reward at time $t \in \{0, 1, 2...\}$ are denoted by the random variables $S_t$, $A_t$, and $R_t$.

The initial state of the environment comes from an initial state distribution $d_0$. Thereafter, the agent follows a policy $\pi$, defined as a conditional distribution over actions given states, i.e., $\pi(a|s) = P(A_t = a|S_t = s)$. The goal of the reinforcement learning agent is to find an optimal policy $\pi^*$ that maximizes the expected sum of discounted accumulated future rewards for a given environment, i.e., $\pi^* \in \arg\max_\pi \mathbb{E}[\sum_{t=0}^{\infty} \gamma^t R_t | \pi]$. For any policy, we also define the state value function $v^\pi(s) = \mathbb{E}[\sum_{k=0}^{\infty} \gamma^k R_{t+k} | \pi, S_t = s]$ and the state-action value function $Q^\pi(s, a) = \mathbb{E}[\sum_{k=0}^{\infty} \gamma^k R_{t+k} | \pi, S_t = s, A_t = a]$.

## 3 RELATED WORK

For the application of state embeddings in reinforcement learning, there are two dominant strands of research, namely *world models* and *state aggregation* using bisimulation metrics. World model approaches train an environment model in a supervised fashion from experience collected in the environment, which is then used to generate compressed state representations (Ha & Schmidhuber, 2018a) or to train an agent using the learned world model (Ha & Schmidhuber, 2018b; Schmidhuber, 2015). Further applications of world models, e.g. for Atari 2000 domains, show that abstract state representations learned via world models can substantially improve sample efficiency (Kaiser et al., 2019; Hafner et al., 2020). Similar to this idea, Munk et al. (2016) pre-train an environment model and use it to provide state representations for an RL agent. Furthermore, de Bruin et al. (2018) investigate additional learning objectives to learn state representations, and Francois-Lavet et al. (2019) propose the use of an environment model to generate abstract state representations; their learned state representations are then used by a Q-learning agent. By using a learned model of the environment to generate abstract states, these approaches capture structure in the state space and reduce the dimensionality of its representation – an idea similar to our proposed embedding model.

In contrast to world models, we considers both states and actions, rather than just states. Additionally, our approach trains an RL agent on the original environment using its embedded representation, rather than learning a policy using a surrogate world model. Bisimulation, on the other hand, is a method for aggregating states that are "behaviorally equivalent" (Li et al., 2006) and can improve convergence speed by grouping similar states into abstract states. Several works on bisimulation-based state aggregation, e.g. Ferns et al. (2004); Givan et al. (2003), present different metrics to measure state similarity. Furthermore, Zhang et al. (2020) and Castro (2020) propose deep learning methods for generating bisimulation-based state aggregations that scale beyond the tabular methods proposed in several earlier works (Castro, 2020). While there are parallels between bisimulation and our approach, we do not propose the aggregation of states. Instead, our embedding technique projects states into a continuous state embedding space, similar to Zhang et al. (2020), where their behavioral similarity is captured in their proximity in embedding space. Furthermore, our method embeds both states and actions and does not employ an explicit similarity metric such as a bisimulation metric, but instead learns the relationships among different states and actions via an environment model. State representations are also used in RL-based NLP tasks, such as Narasimhan et al. (2015), who jointly train an LSTM-based state representation module and a DQN agent, and Ammanabrolu & Riedl (2018), who propose the use of a knowledge graph based state representation.

In addition to state representations, previous work has explored the use of additional models to learn meaningful action representations. In particular, Van Hasselt & Wiering (2009) investigate the use of a continuous actor in a policy gradient based approach to solve discrete action MDPs, where the policy is learned in continuous space and actions are discretized before execution in the environment. Dulac-Arnold et al. (2015) propose a similar methodology, where a policy is learned in continuous action embedding space and then mapped to discrete actions in the original problem domain. Both Van Hasselt & Wiering (2009) and Dulac-Arnold et al. (2015) only consider actions and assume that embeddings are known a priori. Tennenholtz & Mannor (2019) propose a methodology called Act2Vec, where they introduce an embedding model similar to the Skip-Gram model (Mikolov et al., 2013) that is trained using data from expert demonstrations and then combined with a DQN agent. One significant drawback of this approach is that information on the semantics of actions has to be injected via expert demonstration and is not learned automatically. In contrast, Chandak et al. (2019) propose a method that enables the self-supervised learning of action representations from experience collected by the RL agent, using an embedding model that resembles CBOW. The approach presented in (Chandak et al., 2019) is most closely related to our method in the way of integrating embeddings with policy gradient based RL algorithms. However, they only consider action embeddings, while we jointly embed states and actions and improve the performance significantly. As an alternative to embedding methods, Pazis & Parr (2011) and Sallans & Hinton (2004) represent actions in binary format, where they learn a value function associated with each bit, and Sharma et al. (2017) propose the use of factored actions, where actions are expressed using underlying primitive actions. Again, these three methods rely on handcrafted action decomposition.

Previous embedding approaches consider either states or actions in isolation, embedding only one of the two or using separate models for each. By contrast, in addition to capturing structure in the state or action space respectively, we leverage interdependencies between the two by jointly learning embeddings for them, and thus improve the quality of embeddings.

## 4 METHODOLOGY

We propose an embedding model to jointly learn state and action embeddings and present a generic framework that uses the learned embeddings in conjunction with any policy gradient algorithms.

### 4.1 ARCHITECTURE

Our proposed approach has three components: (i) state-action embedding model, (ii) policy gradient based RL agent that takes in the embeded states and outputs actions in action embedding space, and (iii) action mapping function that maps embedded actions to actions in the original problem domain. We define two embedding spaces, i.e., a continuous state embedding space $\mathcal{X} \in \mathbb{R}^m$ and a continuous action embedding space $\mathcal{E} \in \mathbb{R}^d$; states are projected to the state embedding space via a function $\phi : \mathcal{S} \mapsto \mathcal{X}$. On the other hand, we define a function $f : \mathcal{E} \mapsto \mathcal{A}$ that maps points in the embedding space to actions in the original problem domain. With these embedding spaces, we

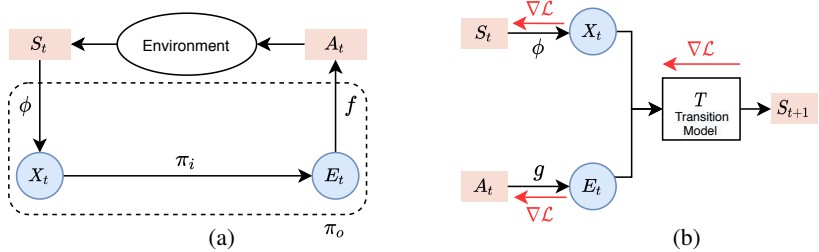

Figure 1: (a) Overall architecture, using the state embedding function $\phi$, the internal policy $\pi_i$, and the action mapping function $f$. (b) Embedding model for learning $\phi$ and the auxiliary action embedding function $g$ (the red arrows denote the gradient flow of the supervised loss in Equation (5)).

further define an internal policy $\pi_i$ that takes in the embedded state and outputs the embedded action, i.e., $\pi_i : \mathcal{X} \mapsto \mathcal{E}$. Furthermore, the mapping functions $\phi$ and $f$, together with the internal policy $\pi_i$ form the overall policy $\pi_o : \mathcal{S} \mapsto \mathcal{A}$, i.e., $\pi_o$ is characterized by the following three components:

$$X_t = \phi(S_t), E_t \sim \pi_i(\cdot|X_t), \text{and } A_t = f(E_t),$$

where random variables $S_t \in \mathcal{S}$, $X_t \in \mathcal{X}$, $E_t \in \mathcal{E}$, and $A_t \in \mathcal{A}$ denote the state in the original problem domain, state in the embedding domain, action in the embedding domain, and action in the original domain at time $t$, respectively. The overall model architecture is illustrated in Figure 1a. In Figure 1a, function $\phi$ is directly learned using an embedding model illustrated in Figure 1b (detailed in Section 4.3.1). However, function $f$ is not directly part of this embedding model, but uses the obtained action embeddings to learn a mapping from $e \in \mathcal{E}$ to $a \in \mathcal{A}$ (see Section 4.3.2 for details).

For an RL problem, the ultimate goal is to optimize the policy $\pi$ in the original problem domain, i.e., $\pi$ in the $\mathcal{S} \times \mathcal{A}$ space. For the proposed architecture, the idea is to first establish efficient embeddings for both states and actions in $\mathcal{S}$ and $\mathcal{A}$. Then we reduce the problem of finding the optimal $\pi^*$ in the original problem domain to the problem of internal policy $\pi_i$ optimization that acts purely in the embedding space. However, it is still unclear how the optimization of $\pi_i$ is related to $\pi^*$. In addition, the overall policy $\pi_o$ relying on $\phi$ and $f$ further complicates the policy learning. In this regard, we establish the relationships among $\pi$, $\pi_o$, and $\pi_i$ in the next section, which is the foundation to justify the validity of leveraging both state-action embeddings in RL.

## 4.2 THEORETICAL FOUNDATIONS: RELATIONSHIPS AMONG $\pi$, $\pi_o$, AND $\pi_i$

To understand the relationships among $\pi$, $\pi_o$, and $\pi_i$, we first derive how $\pi_o$ is related to $\pi$, after which we prove the existence of $\pi_o$ that achieves optimality in the original problem domain. For this goal, two further assumptions are required on the nature of the state embedding function $\phi$ and the action mapping function $f$.

**Assumption 1.** *Given an action embedding $E_t$, $A_t$ is deterministic and defined by a function $f : \mathcal{E} \mapsto \mathcal{A}$, i.e., there exists $a$ such that $P(A_t = a|E_t = e) = 1$.*

**Assumption 2.** *Given a state $S_t$, $X_t$ is deterministic and defined by a function $\phi : \mathcal{S} \mapsto \mathcal{X}$, i.e., there exists $x$ such that $P(X_t = x|S_t = s) = 1$.*

We validate Assumption 1, which defines $f$ as a *many-to-one* mapping, empirically via all experiments conducted in Section 5, and find that no two actions share exactly the same embedded representation, i.e., Assumption 1 holds in practice. Note that we also assume the Markov property for our environment, which is a standard assumption for reinforcement learning problems. With slight abuse of notation, we denote the inverse mapping from an action $a \in \mathcal{A}$ to its corresponding points in the embedding space (*one-to-many* mapping) by $f^{-1}(a) := \{e \in \mathcal{E} : f(e) = a\}$.

**Lemma 1.** *Under Assumptions 1 and 2, for policy $\pi$ in the original problem domain, there exists $\pi_i$ such that*

$$v^\pi(s) = \sum_{a \in A} \int_{\{e\}=f^{-1}(a)} \pi_i(e|x = \phi(s))Q^\pi(s, a)\, de\,.$$

*Proof Sketch.* Based on the Bellman equation for the value function $v^\pi$ in the original domain, we introduce the embedded state $x = \phi(s)$ using Assumption 2. By the law of total probability and Assumption 1, we then introduce the embedded action $e$. From the Markovian property and the definition of our model, we can derive Claims 1 - 5 on (conditional) independence in Appendix A.2, which then allow us to remove the original state and action from the expression of the policy, leaving us with $\pi_i$. See Appendix A.2 for the complete proof. $\qquad\square$

By Assumptions 1 and 2, Lemma 1 allows us to express the overall policy $\pi_o$ in terms of the internal policy $\pi_i$ as

$$\pi_o(a|s) = \int_{\{e\}=f^{-1}(a)} \pi_i(e|x = \phi(s)) \, de \,, \tag{1}$$

bridging the gap between the original problem domain and the policy in embedding spaces $\mathcal{X}$ and $\mathcal{E}$. Under $\pi^*$ in the original domain, we define $v^* := v^{\pi^*}$ and $Q^* := Q^{\pi^*}$ for the ease of discussions. Then using Lemma 1, we now prove the existence of an overall policy $\pi_o$ that is optimal.

**Theorem 1.** *Under Assumptions 1 and 2, there exists an overall policy $\pi_o$ that is optimal, such that $v^{\pi_o} = v^*$.*

*Proof.* Under finite state and action sets, bounded rewards, and $\gamma \in [0, 1)$, at least one optimal policy $\pi^*$ exists. From Lemma 1, we then have

$$v^*(s) = \sum_{a \in A} \int_{\{e\}=f^{-1}(a)} \pi_i(e|\phi(s))Q^*(s, a) \, de \,. \tag{2}$$

Thus, $\exists\, \phi$, $f$, and $\pi_i$, representing an overall policy $\pi_o$, which is optimal, i.e., $v^{\pi_o} = v^*$. $\qquad\square$

Theorem 1 suggests that in order to get the optimal policy $\pi^*$ in the original domain, we can focus on the optimization of the overall policy $\pi_o$, which will be discussed in the next section.

## 4.3 ARCHITECTURE EMBODIMENT AND TRAINING

In this section, we first present the implementation and the joint-training of the state-action embedding model (illustrated in Figure 1b). Based on this method and according to Theorem 1, we then propose a strategy to train the overall policy $\pi_o$, where functions $\phi$ and $f$ are iteratively updated.

### 4.3.1 JOINT TRAINING OF THE STATE-ACTION EMBEDDING

In this section, we elaborate on our proposed embedding model for jointly learning state and action embeddings. Specifically, we require two components: (i) function $\phi$ that projects $S_t$ into embedding space $\mathcal{X}$ and (ii) function $f$ that maps each point in embedding space $\mathcal{E}$ to an action $A_t$ in the original problem domain. Since these functions are not known in advance, we train estimators using a model of the environment (see Figure 1b). This environment model requires two further components: (iii) function $g : \mathcal{A} \mapsto \mathcal{E}$ and (iv) transition model $T : \mathcal{E} \times \mathcal{X} \mapsto \mathcal{S}$ that predicts the next state from the concatenated embeddings of the current state and action. Note that $g$ is a *one-to-one* mapping, since according to Assumption 1, no two actions have exactly the same embedding. Denote the estimators of components (i)-(iv) by $\hat{\phi}, \hat{f}, \hat{g}, \hat{T}$; the target of the environment model is

$$\hat{P}(S_{t+1}|S_t, A_t) = \hat{T}(S_{t+1}|X_t, E_t)\hat{g}(E_t|A_t)\hat{\phi}(X_t|S_t) \,. \tag{3}$$

The difference between the true transition probabilities $P(S_{t+1}|S_t, A_t)$ and the estimated probabilities $\hat{P}(S_{t+1}|S_t, A_t)$ can be measured using the Kullback-Leibler (KL) divergence, where the expectation is over the true distribution $P(S_{t+1}|S_t, A_t)$, i.e.,

$$D_{KL}(P||\hat{P}) = -\mathbb{E}_{S_{t+1}\sim P(S_{t+1}|S_t,A_t)}\left[\ln\left(\frac{\hat{P}(S_{t+1}|S_t, A_t)}{P(S_{t+1}|S_t, A_t)}\right)\right]. \tag{4}$$

Since we observe the full tuple $(S_{t+1}, S_t, A_t)$, we can compute a sample estimate of this using Equation (4). In Equation (4), the denominator does not depend on $\hat{\phi}$, $\hat{g}$, or $\hat{T}$. Therefore, we define the loss function for the embedding model as

$$\mathcal{L}(\hat{\phi}, \hat{g}, \hat{T}) = -\mathbb{E}_{S_{t+1}\sim P(S_{t+1}|S_t,A_t)}\left[\ln(\hat{P}(S_{t+1}|S_t, A_t))\right]. \tag{5}$$

Note that the estimator $\hat{f}$ is not directly included in the embedding model. Instead, we find $\hat{f}$ by minimizing the reconstruction error between the original action and the action reconstructed from the embedding. We can again define this using the KL divergence as

$$\mathcal{L}(\hat{f}) = -\mathbb{E}_{A_t \sim f(A_t|E_t)}\Big[\ln(\hat{f}(A_t|E_t))\Big]. \tag{6}$$

Then, all components of our embedding model can be learned by minimizing the loss functions $\mathcal{L}(\hat{\phi}, \hat{g}, \hat{T})$ and $\mathcal{L}(\hat{f})$. Note that the embedding model is trained in two steps, firstly by minimizing the loss in Equation (5) to update $\hat{\phi}$, $\hat{g}$, and $\hat{T}$ and secondly by minimizing the loss in Equation (6) to update $\hat{f}$. Here, we give the target of the embedding model for the case of discrete state and action spaces. However, the model is equally applicable to continuous domains. In this case, the loss functions $\mathcal{L}(\hat{\phi}, \hat{g}, \hat{T})$ and $\mathcal{L}(\hat{f})$ must be replaced with loss functions suitable for continuous domains. For continuous domains, we adopt a mean squared error loss instead of the losses given in Equations (5) and (6), but other loss functions may also be applicable. For our experiments, we parameterize all components of the embedding model illustrated in Figure 1b as neural networks. More details on this can be found in Appendix A.4.

### 4.3.2 STATE-ACTION EMBEDDING AND POLICY LEARNING

Theorem 1 only shows that optimizing $\pi_o$ can help us achieve the optimality in the original domain. Then a natural question is whether we can optimize $\pi_o$ by directly optimizing the internal policy $\pi_i$ using the state/action embeddings in Section 4.3.1. To answer this question, we first mathematically derive that updating $\pi_i$ is equivalent to updating $\pi_o$. We then proceed to present an iterative algorithm for learning the state-action embeddings and the internal policy.

Suppose $\pi_i$ is parameterized by $\theta$. Then with the objective of optimizing $\pi_o$ by only updating $\pi_i$, we define the performance function of $\pi_o$ as[1]

$$J_o(\phi, \theta, f) = \sum_{s \in \mathcal{S}} d_0(s) \sum_{a \in \mathcal{A}} \pi_o(a|s) Q^{\pi_o}(s, a). \tag{7}$$

Let the state-action-value function for the internal policy be $Q^{\pi_i}(x, e) = \mathbb{E}[\sum_{k=0}^{\infty} \gamma^k R_{t+k} | \pi_i, X_t = x, E_t = e]$. We can then define the performance function of the internal policy as

$$J_i(\theta) = \int_{x \in \mathcal{X}} d_0(x) \int_{e \in \mathcal{E}} \pi_i(e|x) Q^{\pi_i}(x, e) \, de \, dx. \tag{8}$$

The parameters of the overall policy $\pi_o$ can then be learned by updating its parameters $\theta$ in the direction of $\partial J_o(\phi, \theta, f)/\partial\theta$, while the parameters of the internal policy can be learned by updating $\theta$ in the direction of $\partial J_i(\theta)/\partial\theta$. With the following assumption, we then have Lemma 2.

**Assumption 3.** *The state embedding function $\phi(s)$ maps each state $s$ to a unique state embedding $x = \phi(s)$, i.e., $\forall s_i \neq s_j$, $P(\phi(s_i) = \phi(s_j)) = 0$.*

Note that Assumption 3, defining $\phi$ as a *one-to-one* mapping, is not theoretically guaranteed by the definition of our embedding model. Nevertheless, we test this empirically in Section 5 and find that no two states share the same embedded representation in any of our experiments, thus justifying Assumption 3 in practical scenarios considered in this work.

**Lemma 2.** *Under Assumptions 1–3, for all deterministic functions $f$ and $\phi$, which map each point $e \in \mathcal{E}$ and $s \in \mathcal{S}$ to an action $a \in \mathcal{A}$ and to an embedded state $x \in \mathcal{X}$, and the internal policy $\pi_i$ parameterized by $\theta$, the gradient of the internal policy's performance function $\frac{\partial J_i(\theta)}{\partial\theta}$ equals the gradient of the overall policy's performance function $\frac{\partial J_o(\phi, \theta, f)}{\partial\theta}$, i.e.,*

$$\frac{\partial J_i(\theta)}{\partial\theta} = \frac{\partial J_o(\phi, \theta, f)}{\partial\theta}.$$

*Proof Sketch.* Assumptions 1 and 3 allow us to show the equivalence between the internal state-action-value function and the overall state-action-value function. Using this equivalence, we can then remove functions $\phi$ and $f$ from $\partial J_o(\phi, \theta, f)/\partial\theta$ and are left with the gradient of the internal policy $\pi_i$ only. The complete proof is deferred to Appendix A.3. □

---

[1]Equation (7) is for the overall policy with discrete states and actions. Note that our approach is also applicable in continuous domains. $\sum_{s \in \mathcal{S}}$ and $\sum_{a \in \mathcal{A}}$ would then be replaced by $\int_s$ and $\int_a$, respectively.

---

**Algorithm 1:** Joint Training of State-Action Embeddings and the Internal Policy

---

Initialize the state and action embeddings (optional pre-training);
**for** *Epoch = 0, 1, 2 ...* **do**
    **for** *t = 1, 2, 3 ...* **do**
        Generate state embedding $X_t = \phi(S_t)$ ;
        Sample action embedding $E_t \sim \pi_i(\cdot|X_t)$;
        Map embedded action to $A_t = f(E_t)$ ;
        Execute $A_t$ in the environment to observe $S_t, R_t$;
        Update the $\pi_i$ and the critic using some policy gradient algorithms;
        Update $\phi, g, T$, and $f$ by minimizing the losses in Equations (5) and (6);

---

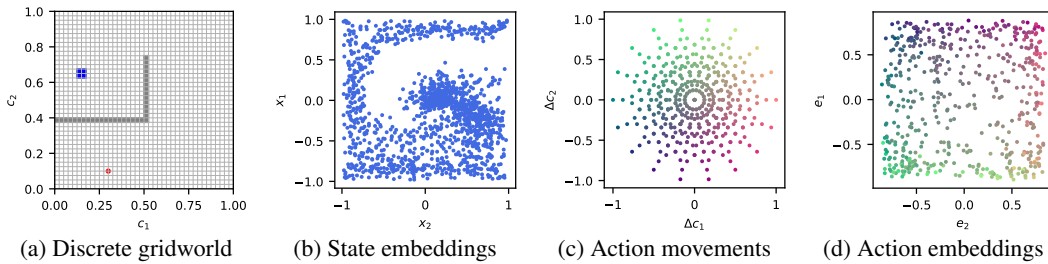

| (a) Discrete gridworld | (b) State embeddings | (c) Action movements | (d) Action embeddings |

Figure 2: State and action embeddings for a discrete state gridworld with $1,600$ states and $2^9$ actions (9 actuators). The embedding dimension for both states and actions is set to 2 for visualization purposes. (a) Original environment, i.e., a discrete gridworld as described in Section 5.1. The blue area is the goal state and the red dot is the starting position of the agent. (b) Learned state embeddings in 2-dimensional state embedding space. (c) Displacement in the Cartesian co-ordinates ($c_1$ and $c_2$) caused by actions. Each action is colored according to this displacement with $[R = \Delta c_1, G = \Delta c_2, B = 0.5]$. (d) Learned action embeddings in continuous action embedding space.

Lemma 2 shows that the updates to the internal policy $\pi_i$ and the overall policy $\pi_o$ are equivalent. This allows us to optimize the overall policy by making updates directly to the internal policy $\pi_i$, thereby avoiding the potentially intractable computation of the inverse functions $f^{-1}$ and $\phi^{-1}$. Since there are no special restrictions on the internal policy $\pi_i$, we can use any policy gradient algorithm designed for continuous control to optimize the policy. In addition, we can also iteratively update parameters for jointly learning $\phi, f$, and $\pi_i$, as shown in Algorithm 1.

## 5    EMPIRICAL EVALUATION

We evaluate our proposed architecture on game-based applications, robotic control, and a real-world recommender system, covering different combinations of discrete and continuous state and action spaces. Our methodology is evaluated using Vanilla Policy Gradient (VPG) (Achiam, 2018), Proximal Policy Optimization (PPO) (Schulman et al., 2017) and Soft Actor Critic (SAC) (Haarnoja et al., 2018) algorithms. We benchmark the performance against these algorithms without embeddings and with action embeddings generated by the method called policy gradients with Representations for Actions (RA) proposed by Chandak et al. (2019). We pre-train the embedding model on randomly collected samples for all experiments and enable continuous updates for the Ant-v2 and recommender system environments. Complete parameterizations are presented in Appendix A.4.

### 5.1    PROOF-OF-CONCEPT: GRIDWORLD AND SLOTMACHINE

*Gridworld:* It is similar to that used by Chandak et al. (2019). States are given as a continuous coordinate, while actions are defined via $n$ actuators, equally spaced around the agent, which move the agent in the direction they are pointing towards. Then each combination of actuators forms an action, resulting in $2^n$ unique actions. We run two sets of experiments in this environment: (i) we use the continuous state directly and (ii) we discretize the coordinate in a $40 \times 40$ grid. The agent receives small step and collision penalties and a large positive reward for reaching the goal state.

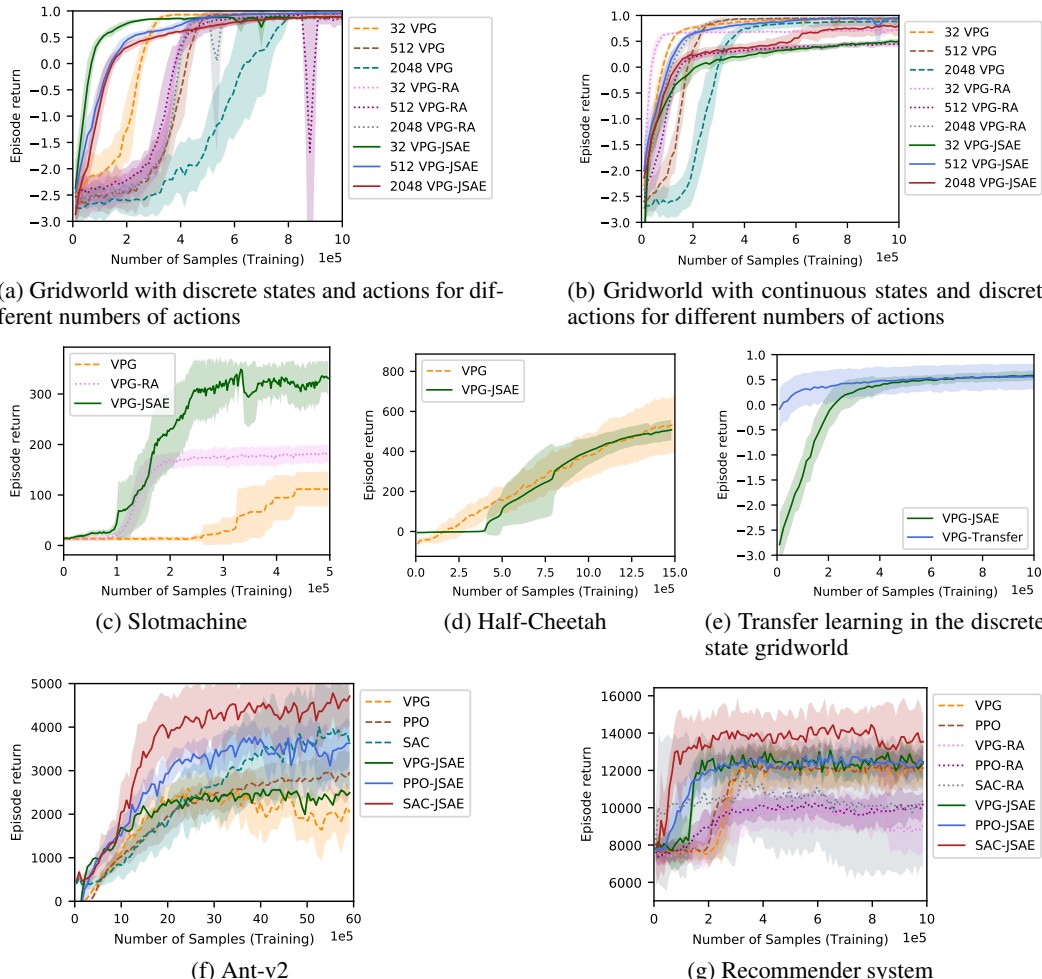

Figure 3: Performance of our approach (JSAE, Joint State Action Embedding) compared against benchmarks without embeddings and/or benchmarks with action embeddings by RA (all results are the average return over 10 episodes, with mean and std. dev. over 10 random seeds).

Figure 2 illustrates the learned embeddings for this environment. The obtained embeddings for states and actions illustrate that our approach is indeed able to obtain meaningful representations. Actions are embedded according to the displacement in the Cartesian coordinates they represent (Figure 2d). Figure 2b shows that states are embedded according to the coordinate they represent and that the embeddings capture the L-shaped obstacle in the original problem domain.

*Slotmachine:* It consists of several reels, each of which can be individually spun by the agent for a fraction of a full turn. Then, each unique combination of these fractions constitutes an action. The discrete state is given by the numbers on the front of the reels.

Both scenarios yield large discrete action spaces, rendering them well-suited for evaluating our approach. The results for the gridworld and slotmachine environments are shown in Figure 3a, 3b, and 3c. Our approach outperforms both benchmarks on the discretized gridworld and the slotmachine environments and is comparable for the continuous-state gridworld. Such observation suggests that our approach is particularly well suited for discrete domains. Intuitively, this is because the relationship between different states or action in discrete domains is usually not apparent, e.g., in one-hot encoding. In continuous domains, however, structure in the state and action space is already captured to some extent, rendering it harder to uncover additional structure by embeddings; nevertheless, when a continuous problem becomes more complicated, our embedding method again shows effectiveness in capturing useful information (see Section 5.2).

## 5.2 ROBOTIC CONTROL

We use the *half-cheetah-v2* and the *Ant-v2* environments from the robotic control simulator Mujoco (Todorov et al., 2012). Here, the agent observes continuous states (describing the current position of the cheetah or ant respectively) and then decides on the force to apply to each of the joints of the robot (i.e., continuous actions). Rewards are calculated based on the distance covered in an episode. From the results presented in Figure 3d, we observe that our method does not outperform the benchmark on the half-cheetah-v2 environment. On the Ant-v2 environment, however, our approach outperforms the benchmarks. Similar to the experiments in a continuous gridworld, it is less likely that embedding uncovers additional structure on top of the structure inherent to the continuous state and action, explaining the comparable performance with and without embeddings on the half-cheetah-v2 domain. Nevertheless, in the more challenging Ant-v2 environment, the state and action representation is more complex and the agent might thus benefit from the reduced dimensionality in embedding space. Additionally, the learned embeddings appear to be able to capture sufficient structure in the state and action space, despite using an embedding space with lower dimensionality.

## 5.3 RECOMMENDER SYSTEM

In addition, we also test our methodology on a real-world application of a recommender system. We use data on user behavior from an e-commerce store collected in 2019 (Kechinov, 2019). The environment is constructed as an $n$-gram based model, following Shani et al. (2005). Based on the $n$ previously purchased items, a user's purchasing likelihood is computed for each item in the store. Recommending an item then scales the likelihood for that item by a pre-defined factor. States are represented as a concatenation of the last $n$ purchased items and each item forms an action. In this environment we have $835$ items (actions) and approx. $700,000$ states. The results obtained under various RL methods are reported in Figure 3g. We find that our approach leads to significant improvements in both the convergence speed and final performance compared to benchmarks. This result confirms that our method is particularly useful in the presence of large discrete state and action spaces. Interestingly, the PPO and VPG benchmarks outperform the benchmarks using the RA methodology. We conjecture that the action embeddings generated in RA on this environment in fact obfuscate the effect an action has, and thus limit the agent's performance.

## 5.4 APPLICATION IN TRANSFER LEARNING

In addition to improving the convergence speed and the accumulated reward, we test whether the policies learned in one environment can be transferred to a similar one. We consider two environments (old and new) that differ by only the state space or the action space. To leverage the previous experience, if the two environments have the same state (or action) space, the state (or action) embedding component in the new environment is initialized with the weights learned from the old environment; we then train our model in this new environment. For evaluation, we use two gridworlds, where only the numbers of actions differ, i.e., 512 (old) and 2048 (new). Results in Figure 3e show that the model with the transferred knowledge outperforms the case where the entire model is trained from scratch, in terms of the convergence speed. Therefore, it confirms that the previously learned policy serves as a critical initialization for the new gridworld environment and sheds light on the potential of our approach in transfer learning; further evaluations are left for future work.

## 6 CONCLUSION

In this paper, we presented a new architecture for jointly training states/action embeddings and combined this with common reinforcement learning algorithms. Our theoretical results confirm the validity of the proposed approach, i.e., the existence of an optimal policy in the embedding space. We empirically evaluated our method on several environments, where it outperforms state-of-the-art RL approaches in complex large-scale problems. Our approach is easily extensible as it can be combined with most existing RL algorithms and there are no special restrictions on the parameterizations of the embedding model. Consequently, we conclude that the proposed methodology works towards the applicability of reinforcement learning to real-world problems.

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

## A APPENDIX

### A.1 CLAIMS

In addition to Assumptions 1 and 2, we derive three claims on conditional independence from the definition of our embedding model and the Markovian property of the environment, listed below.

**Claim 1.** *Action $A_t$ is conditionally independent of $S_t$ given a state embedding $X_t$:*

$$P(A_t = a, S_t = s | X_t = x) = P(A_t = a | X_t = x) P(S_t = s | X_t = x).$$

**Claim 2.** *Action $A_t$ is conditionally independent of $S_t$ given state and action embeddings $X_t$ and $E_t$:*

$$P(A_t = a, S_t = s | X_t = x, E_t = e) = P(A_t = a | X_t = x, E_t = e) P(S_t = s | X_t = x, E_t = e).$$

**Claim 3.** *Next state $S_{t+1}$ is conditionally independent of $E_t$ and $X_t$ given $A_t$ and $S_t$:*

$$P(S_{t+1} = s', E_t = e, X_t = x | A_t = a, S_t = s) =$$
$$P(S_{t+1} = s' | A_t = a, S_t = s) P(E_t = e, X_t = x | A_t = a, S_t = s).$$

Using Claims 1 - 3, we can derive two further auxiliary claims that will be used in the proof of Lemma 1.

**Claim 4.** *The probability of next state $S_{t+1}$ is independent of $E_t$ and $X_t$, given state $S_t$ and action $A_t$, i.e.,*

$$P(S_{t+1} = s' | E_t = e, X_t = x, A_t = a, S_t = s) = P(S_{t+1} = s' | A_t = a, S_t = s).$$

*Proof.* From Claim 3, we have

$$P(s', e, x | a, s) = P(s' | a, s) P(e, x | a, s)$$
$$P(s' | e, x, a, s) P(e, x | a, s) = P(s' | a, s) P(e, x | a, s)$$
$$P(s' | e, x, a, s) = P(s' | a, s).$$

$\square$

**Claim 5.** *The probability of action $A_t$ is independent of $S_t$ and $X_t$, given action embedding $E_t$, i.e.,*

$$P(A_t = a | S_t = s, X_t = x, E_t = e) = P(A_t = s | E_t = e).$$

*Proof.* From Claim 2, we have

$$P(a, s | x, e) = P(a | x, e) P(s | x, e)$$
$$P(a | s, x, e) = P(a | x, e).$$

Since action $a$ only depends on the action embedding $e$ in our model, this becomes

$$P(a | s, x, e) = P(a | e).$$

$\square$

### A.2 PROOF OF LEMMA 1

**Lemma 1.** *Under Assumptions 1 and 2, for policy $\pi$ in the original problem domain, there exists $\pi_i$ such that*

$$v^\pi(s) = \sum_{a \in \mathcal{A}} \int_{\{e\} = f^{-1}(a)} \pi_i(e | x = \phi(s)) Q^\pi(s, a)\, de$$

*Proof.* The Bellman equation for a MDP is given by

$$v^\pi(s) = \sum_{a \in \mathcal{A}} \pi(a|s) \sum_{s' \in \mathcal{S}} P(s'|s,a)G \,,$$

where $G$ denotes the return, i.e. $[\mathcal{R}(s,a) + \gamma v^\pi(s')]$, which is a function of $s, a$, and $s'$. By re-arranging terms we get

$$
\begin{aligned}
v^\pi(s) &= \sum_{a \in \mathcal{A}} \sum_{s' \in \mathcal{S}} \pi(a|s) P(s'|s,a)G \\
&= \sum_{a \in \mathcal{A}} \sum_{s' \in \mathcal{S}} \pi(a|s) \frac{P(s',s,a)}{P(s,a)}G \\
&= \sum_{a \in \mathcal{A}} \sum_{s' \in \mathcal{S}} \pi(a|s) \frac{P(s',s,a)}{\pi(a|s)P(s)}G \\
&= \sum_{a \in \mathcal{A}} \sum_{s' \in \mathcal{S}} \frac{P(a|s',s)P(s',s)}{P(s)}G \,.
\end{aligned}
$$

Since $s$ can be deterministically mapped to $x$ via $x = \phi(s)$, by Assumption 2, we have

$$
\begin{aligned}
v^\pi(s) &= \sum_{a \in \mathcal{A}} \sum_{s' \in \mathcal{S}} \frac{P(x,a|s',s)P(s',s)}{P(s)}G \\
&= \sum_{a \in \mathcal{A}} \sum_{s' \in \mathcal{S}} \frac{P(x,a|s',s)P(s',s)P(x|s)}{P(x|s)P(s)}G \\
&= \sum_{a \in \mathcal{A}} \sum_{s' \in \mathcal{S}} \frac{P(x,a,s',s)P(x|s)}{P(x,s)}G \\
&= \sum_{a \in \mathcal{A}} P(x|s) \sum_{s' \in \mathcal{S}} \frac{P(x,a,s',s)}{P(x,s)}G \\
&= \sum_{a \in \mathcal{A}} P(x|s) \sum_{s' \in \mathcal{S}} \frac{P(a,s'|x,s)P(x,s)}{P(x,s)}G \\
&= \sum_{a \in \mathcal{A}} P(x|s) \sum_{s' \in \mathcal{S}} P(a,s'|x,s)G \\
&= \sum_{a \in \mathcal{A}} P(x|s) \sum_{s' \in \mathcal{S}} P(s'|a,x,s)P(a|s,x)G \,.
\end{aligned}
$$

From Claim 1, we know that $P(a|s,x) = P(a|x)$. Therefore,

$$v^\pi(s) = \sum_{a \in \mathcal{A}} P(x|s) \sum_{s' \in \mathcal{S}} P(s'|a,x,s)P(a|x)G \,.$$

Since $P(x|s)$ is deterministic by Assumption 2 and evaluates to 1 for the representation of $\phi(s) = x$, we can rewrite the equation above to

$$v^\pi(s) = \sum_{a \in \mathcal{A}} \sum_{s' \in \mathcal{S}} P(s'|a,x,s)P(a|x)G \,.$$

We now proceed to establish the relationship with the action embedding. From above we have

$$v^{\pi}(s) = \sum_{a \in \mathcal{A}} \sum_{s' \in \mathcal{S}} P(a|x)P(s'|a,x,s)G$$

$$= \sum_{a \in \mathcal{A}} \sum_{s' \in \mathcal{S}} P(a|x)\frac{P(s',a,x,s)}{P(a,x,s)}G \,.$$

From Claim 1, $a$ and $s$ are conditionally independent given $x$. Therefore, $P(a,x,s) = P(a|x)P(s|x)P(x)$, which allows us to rewrite the above equation as

$$v^{\pi}(s) = \sum_{a \in \mathcal{A}} \sum_{s' \in \mathcal{S}} P(a|x)\frac{P(s',a,x,s)}{P(a|x)P(x,s)}G$$

$$= \sum_{a \in \mathcal{A}} \sum_{s' \in \mathcal{S}} \frac{P(s',a,x,s)}{P(x,s)}G$$

$$= \sum_{a \in \mathcal{A}} \sum_{s' \in \mathcal{S}} \frac{P(a|s',x,s)P(s',x,s)}{P(x,s)}G \,.$$

By the law of total probability, we can now introduce the new variable $e$, which is the embedded action. Then

$$v^{\pi}(s) = \sum_{a \in \mathcal{A}} \sum_{s' \in \mathcal{S}} \int_e \frac{P(a,e|s',x,s)P(s',x,s)}{P(x,s)}G\,de$$

$$= \sum_{a \in \mathcal{A}} \sum_{s' \in \mathcal{S}} \int_e P(e|x,s)\frac{P(a,e|s',x,s)P(s',x,s)}{P(e|x,s)P(x,s)}G\,de \,.$$

Since $e$ is uniquely determined by $x$, we can drop $s$ in $P(e|x,s)$, and thus

$$v^{\pi}(s) = \sum_{a \in \mathcal{A}} \sum_{s' \in \mathcal{S}} \int_e P(e|x)\frac{P(a,e,s',x,s)}{P(e,x,s)}G\,de$$

$$= \sum_{a \in \mathcal{A}} \sum_{s' \in \mathcal{S}} \int_e P(e|x)P(a,s'|e,x,s)G\,de$$

$$= \sum_{a \in \mathcal{A}} \sum_{s' \in \mathcal{S}} \int_e P(e|x)P(s'|a,e,x,s)P(a|x,e,s)G\,de \,.$$

Using the previously derived Claims 4 and 5, the above equation can be simplified to

$$v^{\pi}(s) = \sum_{a \in \mathcal{A}} \sum_{s' \in \mathcal{S}} \int_e P(e|\phi(s))P(s'|a,s)P(a|e)G\,de \,.$$

Since the function $f$, mapping $e$ to $a$, is deterministic by Assumption 1 and only evaluates to 1 for a particular $a$ and 0 elsewhere, we can rewrite this further as

$$v^{\pi}(s) = \sum_{a \in \mathcal{A}} \sum_{s' \in \mathcal{S}} \int_{f^{-1}(a)} P(e|\phi(s))P(s'|a,s)G\,de \,.$$

Summarizing the terms, this becomes

$$v^{\pi}(s) = \sum_{a \in \mathcal{A}} \int_{f^{-1}(a)} \pi_i(e|\phi(s))Q^{\pi}(a,s)\,de \,.$$

$\square$

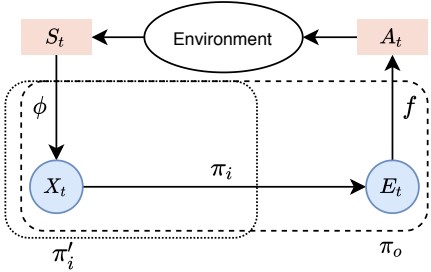

Figure 4: Illustration of $\pi_i'$ in the combined model.

## A.3 PROOF OF LEMMA 2

**Lemma 2.** *Under Assumptions 1–3, for all deterministic functions $f$ and $\phi$ which map each point $e \in \mathcal{E}$ and $s \in \mathcal{S}$ to an action $a \in \mathcal{A}$ and to an embedded state $x \in \mathcal{X}$, and the internal policy $\pi_i$ parameterized by $\theta$, the gradient of the internal policy's performance function $\frac{\partial J_i(\theta)}{\partial \theta}$ equals the gradient of the overall policy's performance function $\frac{\partial J_o(\phi, \theta, f)}{\partial \theta}$, i.e.,*

$$\frac{\partial J_i(\theta)}{\partial \theta} = \frac{\partial J_o(\phi, \theta, f)}{\partial \theta} \,.$$

*Proof.* Recall from Lemma 1 that the overall policy is defined using the internal policy

$$\pi_o(a|s) = \int_{f^{-1}(a)} \pi_i(e|\phi(s)) \, de \,.$$

We can then define the performance function of the overall policy using the internal policy as

$$J_o(\phi, \theta, f) = \sum_{s \in \mathcal{S}} d_0(s) \sum_{a \in \mathcal{A}} \int_{f^{-1}(a)} \pi_i(e|\phi(s)) Q^{\pi_o}(s, a) \, de \,.$$

The gradient of this performance function w.r.t. $\theta$ is

$$\frac{\partial J_o(\phi, \theta, f)}{\partial \theta} = \frac{\partial}{\partial \theta} \Big[ \sum_{s \in \mathcal{S}} d_0(s) \sum_{a \in \mathcal{A}} \int_{f^{-1}(a)} \pi_i(e|\phi(s)) Q^{\pi_o}(s, a) \, de \Big] \,. \tag{9}$$

Now consider another policy $\pi_i'$ as illustrated in Figure 4, i.e., a policy that takes the raw state input $S_t$ and outputs $E_t$ in embedding space. By Assumption 1, $Q^{\pi_o}(s, a) = Q^{\pi_i'}(s, e)$ as $e$ is deterministically mapped to $a$. However, $Q^{\pi_i}(x, e) = \mathbb{E}_{s \sim \phi^{-1}(x)}[Q^{\pi_i'}(s, e)] = \mathbb{E}_{s \sim \phi^{-1}(x)}[Q^{\pi_o}(s, a)]$, where $\phi^{-1}(\cdot)$ is the inverse mapping from $x$ to $s$. Nevertheless, by Assumption 3, $\mathbb{E}_{s \sim \phi^{-1}(x)}[Q^{\pi_o}(s, a)] = Q^{\pi_o}(s, a)$. Therefore, we can replace the overall $Q^{\pi_o}(a, s)$ in Equation (9) by the internal $Q^{\pi_i}(\phi(s), e)$ as follows

$$\frac{\partial J_o(\phi, \theta, f)}{\partial \theta} = \frac{\partial}{\partial \theta} \Big[ \sum_{s \in \mathcal{S}} d_0(s) \sum_{a \in \mathcal{A}} \int_{f^{-1}(a)} (\pi_i(e|\phi(s)) Q^{\pi_i}(\phi(s), e) \Big] \, de \,.$$

Then the deterministic mapping of $e$ to $a$ by function $f$ further allows us to replace the integral over $f^{-1}(a)$ and the summation over $\mathcal{A}$ with an integral over $e$. Moreover, by Assumption 3, we have $d_0(s) = d_0(x = \phi(s))$. Therefore, the above can be rewritten as

$$\frac{\partial J_o(\phi, \theta, f)}{\partial \theta} = \frac{\partial}{\partial \theta} \Big[ \int_{x \in \mathcal{X}} d_0(x) \int_{e \in \mathcal{E}} \pi_i(e|x) Q^{\pi_i}(x, e) \, de \, dx \Big]$$

$$= \frac{\partial J_i(\theta)}{\partial \theta} \,.$$

$\square$

## A.4 PARAMETERIZATION

### A.4.1 IMPLEMENTATION AND ENVIRONMENTS

We implement all evaluated models in Python and use the PyTorch library for neural network implementation (Paszke et al., 2019). The implementation for the benchmark models is adapted from OpenAI's SpinningUp package (Achiam, 2018). In the following, we elaborate on the different environments used for experimentation.

**Gridworld** The gridworld environments are constructed using a $1 \times 1$ grid to represent the underlying continuous state space. For the continuous state version of the environment, a continuous coordinate with the position of the agent in this grid is used as the state. In the discrete state version, we discretize this continuous coordinate in a $40 \times 40$ grid. Actions in these environments are generated from $n$ actuators, equally spaced around the agent, which move the agent in the direction they are pointing towards. Then each combination of actuators forms a unique action, resulting in $2^n$ unique actions. An illustration of this environment can be seen in Figure 5. We limit the maximum number of steps the agent can take in an episode to $150$ and set the maximum step size to $0.2$. The agent receives a positive reward of $1$ for reaching the goal step and receives a small penalty of $-0.01$ for each step taken and an additional penalty of $-0.05$ for colliding with an obstacle or attempting to leave the grid.

**Slotmachine** The slotmachine environment is defined using $4$ reels, with $6$ values per reel, randomly sampled from the range between $0$ and $8$. Note that we sample these reels once and then keep them fixed across all experiments to ensure that these run in the same environment. The action space is defined by the fraction of a full turn for which each reel is spun. We set the maximum fraction to $0.8$ and use increments of $0.1$ to set the fraction of a full turn. Since each combination of fractions forms an action, we get $4,096$ actions in total. The maximum number of steps per episode is set to $20$. Rewards are calculated according to

$$R_t = \sum_{i=1,2,3,4} v_i \times \mathbb{1}\{n_i \geq m\},$$

where $v_i$ denotes the value of the $i^{th}$ reel, $n_i$ is the number of times $v_i$ occurred across all $4$ reels, $m$ is the minimum number of times it has to occur to result in non-zero reward, and $\mathbb{1}\{\cdot\}$ is an indicator function. In our experiments, we set the threshold $m = 3$.

**Robotic Control: Half-Cheetah and Ant-v2** For the experiments in a robotic control environment, we use the physics engine Mujoco and use the *Half-Cheetah* and *Ant-v2* environments implemented therein (Todorov et al., 2012). We limit the maximum number of steps per episode to $250$ for the Half-Cheetah environment and to $1000$ for the Ant-v2 environment.

**Real-world Recommender System** The environment model for the recommender system is constructed from a publicly available data set containing data on approximately $43$ million user interactions collected from a multi-category e-commerce store (Kechinov, 2019). We only use purchase

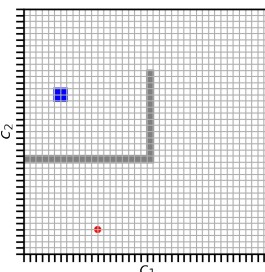

Figure 5: Gridworld environment as used for evaluation The goal state is in blue, the agent is in red, and the obstacles are shown in gray.

events to create our environment model, where we filter out items that were bought fewer than $100$ times and users that made less than $3$ purchases. We are then left with approximately $250,000$ data points and $835$ items. The environment model is then created using an $n$-gram based model, with $n$ set to $2$. The state is represented as the concatenation of the $n$ last items purchased and each item available for recommendation forms an action. We thus have $835$ actions and $697,225$ possible states. In the environment model, we compute the purchasing probability for a user for all items, given the current state, i.e., the $n$ last purchases that user made. We parameterize a categorical distribution with these transition probabilities to sample the next item. The RL agent choses which item to recommend. Recommending an item increases its transition probability by a factor $\alpha = 3$, after which the sum of all transition probabilities for the current state are normalized to sum to $1$. The transition function in this environment is stochastic as we have transition probabilities for each item and each state. For more details on the environment model used, please refer to Shani et al. (2005), whose methodology is adopted. As the reward function, we simply use the revenue obtained in each time step, i.e., the price of the item that the customer bought after making a certain recommendation.

### A.4.2 Hyper Parameters

Training for all experiments is conducted in epochs, where each epoch corresponds to a set of samples. At the end of an epoch, the actor and critic are trained for a number of iterations using their respective learning rates. For the VPG and PPO algorithms, the actor network is trained for $3$ iterations and the critic network is trained for $6$ iterations at the end of an epoch in all of the conducted experiments, using a single batch in each iteration, consisting of all samples collected during the epoch. The SAC updates are also run at the end of an epoch, where we train the actor and critic on $i$ batches sampled from the replay buffer, where $i$ is set equal to the number of samples collected during the epoch. For all experiments, training is conducted in epochs, where a fixed number of samples are collected per epoch. Rather than updating the models after every timestep or episode, we run several iterations of gradient descent at the end of an epoch (see the reference implementations in Achiam (2018)). The batch size is searched over $\{256, 512\}$ in all SAC experiments. The environment configuration is kept fixed for each environment across different models as well as the hyper parameter searches to ensure comparability.

We conduct a gridsearch over different parameter combinations for each experiment and report the average per episode return and standard deviation achieved over $10$ runs with different random seeds, using the hyper parameters that perform the best in the gridsearch. The best hyper parameters – apart from those we kept fixed – are reported in Tables 1 - 7. The actor and critic learning rates are searched over $\{0.001, 0.0003, 0.0001\}$ for all environments. We also search the initialization value for the standard deviation of the actor over $\{0.3, 0.6, 0.8, 1.0\}$ in all experiments and keep $\gamma = 0.99$ and $\lambda = 0.97$ fixed. Note that $\lambda$ is only used for general advantage estimation in the VPG algorithm. For all experiments apart from those using the recommender system and the Ant-v2 environment, we pre-train the embedding model using $30,000$ randomly collected samples and keep the embeddings fixed thereafter. For the recommender system, we pre-train the embedding model using $50,000$ randomly collected samples andalso allow continuous updating of the learned embeddings. Similarly, for the Ant-v2 environment, we pre-train the embedding model using $100,000$ randomly collected samples and then enable continuous updating. In all cases, we apply a *Tanh* non-linearity to the learned embeddings to bound their effective range. Throughout our experiments, we keep the learning rate of the embedding model fixed at $0.01$. For continuous domains, we parameterize the mapping function $f$ as a neural network with a single layer with $64$ hidden units. In discrete domains, we use a simpler nearest neighbor mapping function to parameterize $f$, i.e., we map the output of the internal policy $\pi_i$ to the action that is closest in embedding space $\mathcal{E}$.

For experiments on the gridworld environments, we fix the state and action embedding dimensions at $8$ and $2$, respectively. The network size for the actor and critic is set to $(64, 64)$, i.e., a two-layer network with $64$ neurons in each layer. We then train the agent for $700$ epochs, with $1,500$ steps, i.e., a minimum of $10$ episodes per epoch.

In the experiments on the slotmachine environment, the dimension for state and action embeddings is searched over $\{8, 16\}$ and $\{2, 4\}$, respectively. The network size for the actor and critic is kept fixed at $(64, 64)$. The agent is trained for $2,500$ epochs, with $200$ samples (a minimum of $10$ episodes per epoch).

| Parameter | VPG (5) discrete | VPG (9) disc. | VPG (11) disc. | VPG (5) cont. | VPG (9) cont. | VPG (11) cont. |
|---|---|---|---|---|---|---|
| Critic Learning Rate | 0.001 | 0.0003 | 0.001 | 0.001 | 0.001 | 0.0001 |
| Actor Learning Rate | 0.001 | 0.001 | 0.001 | 0.001 | 0.001 | 0.001 |
| Actor Std. Dev. | – | – | – | 0.8 | 0.6 | 0.8 |

Table 1: Best hyper parameters for the gridworld experiments without embedding.

| Parameter | VPG-JSAE (5) disc. | VPG-JSAE (9) disc | VPG-JSAE (11) disc. | VPG-JSAE (5) cont. | VPG-JSAE (9) cont. | VPG-JSAE (11) cont. |
|---|---|---|---|---|---|---|
| Critic Learning Rate | 0.0003 | 0.001 | 0.001 | 0.0003 | 0.001 | 0.001 |
| Actor Learning Rate | 0.0003 | 0.0001 | 0.0003 | 0.0001 | 0.0003 | 0.0003 |
| Actor Std. Dev. | 0.8 | 1.0 | 1.0 | 0.3 | 1.0 | 1.0 |

Table 2: Best hyper parameters for the gridworld experiments with embedding (JSAE).

For experiments on the Half-Cheetah environment, the state and action embedding dimensions are searched over $\{10, 20\}$ and $\{6, 10\}$, respectively. We search the network size over $\{(64, 64), (128, 128)\}$ and train the agent for $1,000$ epochs with $1,500$ steps per epoch, i.e., at least 6 episodes per epoch.

In the experiments on the Ant-v2 environment, the dimensions of the state and action embeddings are searched over $\{16, 32, 64\}$ and $\{6, 8, 10\}$, respectively. We search the network size over $\{(64, 64), (128, 128)\}$ and train the agent for $2,000$ epochs with $3,000$ steps per epoch, i.e., a minimum of 3 episodes per epoch. The embeddings are pre-trained using $100,000$ randomly generated samples. Continuous updating for the leared embedding model is enabled for the Ant-v2 environment.

On the recommender system environment, we evaluate Proximal Policy Optimisation (PPO) and Soft Actor Critic (SAC) algorithms in addition to the previously used Vanilla Policy Gradient (VPG). For the experiments using the SAC algorithm, we search the actor parameter $\alpha$, which corresponds to an exploration-exploitation trade-off over $\{0.1, 0.3\}$. For experiments involving a PPO algorithm, we set the gradient clipping ratio to $0.2$ and keep this fixed throughout. Embedding dimensions for states and actions are searched over $\{8, 16, 32\}$ and $\{4, 8, 12\}$, respectively. In all experiments, the embedding functions are pre-trained using $200,000$ randomly generated samples. In contrast to experiments on other environments, we enable continuous updating of these learned embeddings for the recommender system. We search the size of the actor and critic network over $\{(64, 64), (128, 128)\}$ and train the agent for $1,000$ epochs with $1,000$ steps per epoch. The episode length is set to 20 steps.

| Parameter | VPG-RA (5) disc. | VPG-RA (9) disc. | VPG-RA (11) disc. | VPG-RA (5) cont. | VPG-RA (9) cont. | VPG-RA (11) cont. |
|---|---|---|---|---|---|---|
| Critic Learning Rate | 0.0003 | 0.001 | 0.0003 | 0.0001 | 0.0003 | 0.0001 |
| Actor Learning Rate | 0.0003 | 0.0003 | 0.0003 | 0.001 | 0.0003 | 0.0003 |
| Actor Std. Dev. | 0.8 | 0.6 | 0.6 | 0.8 | 0.8 | 1.0 |

Table 3: Best hyper parameters for the gridworld experiments with embedding (RA).

| Parameter | VPG | VPG-JSAE | VPG-RA |
|---|---|---|---|
| Critic Learning Rate | 0.001 | 0.0001 | 0.001 |
| Actor Learning Rate | 0.0003 | 0.0003 | 0.0003 |
| State Embedding Dim. | – | 16 | – |
| Action Embedding Dim. | – | 4 | 2 |
| Actor Std. Dev. | – | 0.6 | 1.0 |

Table 4: Best hyper parameters for the slotmachine experiments.

| Parameter | VPG | VPG-JSAE |
|---|---|---|
| Critic Learning Rate | 0.0003 | 0.001 |
| Actor Learning Rate | 0.0003 | 0.0003 |
| State Embedding Dim. | – | 20 |
| Action Embedding Dim. | – | 10 |
| Actor Std. Dev. | 0.6 | 0.6 |
| Actor Hidden Units | (128, 128) | (128, 128) |
| Critic Hidden Units | (128, 128) | (128, 128) |

Table 5: Best hyper parameters for the Half-Cheetah experiments.

| Parameter | VPG | PPO | SAC | VPG-JSAE | PPO-JSAE | SAC-JSAE |
|---|---|---|---|---|---|---|
| Critic Learning Rate | 0.001 | 0.0003 | 0.001 | 0.001 | 0.001 | 0.001 |
| Actor Learning Rate | 0.0001 | 0.0003 | 0.001 | 0.001 | 0.001 | 0.001 |
| State Embedding Dim. | – | – | – | 64 | 32 | 64 |
| Action Embedding Dim. | – | – | – | 8 | 8 | 8 |
| Alpha | – | – | 0.1 | – | – | 0.1 |
| Actor Std. Dev. | 0.6 | 0.8 | 0.6 | 0.6 | 0.8 | 0.8 |
| Batch Size | – | – | – | – | – | 256 |
| Actor Hidden Dim. | (128, 128) | (128, 128) | (128, 128) | (128, 128) | (128, 128) | (128, 128) |
| Critic Hidden Dim. | (128, 128) | (128, 128) | (128, 128) | (128, 128) | (128, 128) | (128, 128) |

Table 6: Best hyper parameters for the Ant-v2 experiments.

| Parameter | VPG | PPO | VPG-JSAE | PPO-JSAE | SAC-JSAE | VPG-RA | PPO-RA | SAC-RA |
|---|---|---|---|---|---|---|---|---|
| Critic Learning Rate | 0.001 | 0.0001 | 0.0003 | 0.001 | 0.001 | 0.0001 | 0.0001 | 0.0001 |
| Actor Learning Rate | 0.001 | 0.001 | 0.001 | 0.001 | 0.001 | 0.001 | 0.001 | 0.0001 |
| Alpha | – | – | – | – | 0.1 | – | | 0.1 |
| State Embedding Dim. | – | – | 8 | 8 | 8 | – | – | – |
| Action Embedding Dim. | – | – | 8 | 8 | 8 | 8 | 8 | 8 |
| Actor Std. Dev. | – | – | 0.8 | 0.8 | 0.8 | 0.6 | 0.6 | 0.6 |
| Batch Size | – | – | – | – | 256 | – | – | 256 |
| Actor Hidden Dim. | (64, 64) | (64, 64) | (64, 64) | (64, 64) | (64, 64) | (64, 64) | (64, 64) | (128, 128) |
| Critic Hidden Dim. | (64, 64) | (64, 64) | (64, 64) | (64, 64) | (64, 64) | (64, 64) | (64, 64) | (128, 128) |

Table 7: Best hyper parameters for the recommender system experiments.

