# OpenReview forum: "Jointly-Trained State-Action Embedding for Efficient Reinforcement Learning"
_ICLR.cc/2021/Conference — Reject_

### Official Review · AnonReviewer3 · 2020-10-28
**The paper presents a somewhat novel way to learn and to use joint state-action representation for policy gradient methods in a deep learning setting. There are some concerns about the soundness of the method.**

**Rating:** 5
**Confidence:** 4

**Review:**

The paper presents a method to take advantage of joint state-action representation. It proposes to use an environment model to obtain embeddings for state and action and then such representation should enable better generalization across state-action space. Experiments on several gaming and recommendation systems are conducted to show the superior performance of the proposed method.

Novelty. The idea is not new but the particular method of defining a policy in the embedding space seems to be new.

Significance. The work may be of general interest to the reinforcement learning research community.

Clarity. The paper presents the key idea/method clearly.

Quality. There are a few theoretical and empirical issues in this work.

First, one action can correspond to many embeddings and one embedding can correspond to only one action. Shouldn’t it be the case that one embedding can correspond to multiple actions? Intuitively, one wants to enjoy generalization across different actions. And generalization ability is also one of the goals mentioned by the authors.

Second, I do not see why the assumptions can get satisfied in practice.

Third, based on the two assumptions, lemma 1 and theorem 1 are not that interesting. Furthermore, the bottom line says that theorem 1 indicates one can focus on the optimization of the overall policy \pi_0; however, theorem 1 indicates only the existence of \pi_0 equal to the optimal policy in the original MDP, it is unclear if optimizing \pi_0 according to Alg 1 can really lead to the optimal one.

Motivation. I think this is another serious concern of this paper. Although I agree with the general direction of learning some sort of joint embedding of state-action pairs, I am not persuaded by the motivation of using the method proposed by the authors. Many works are indeed using state-action representation, especially on those continuous control problems. Whenever taking both state and action as input for the critic/value network, one can think of the final hidden layer as the learned joint state-action representation. Why such simple way cannot be used? At least, those should be compared in the experiments.

Experiments. In general, the experimental results are not really strong. Only Fig 2(a)(c) show a clear advantage of using joint representation. Given the weak motivation of the proposed method and the proposed theory, I think the most important experiment should be designed to show the method is indeed superior to other intuitive baselines. As I mentioned, one baseline is to simply take both state-action as input. Another intuitive baseline is to simply learn state-action representation by using an environment model first and then based on the learned representation to learn a policy. To enable the separability of the state and action representations, one can use the pairwise product to generate the joint representation.

---

> ### Author Response · Authors · 2020-11-24
> **Response: Comments and Questions**
>
> Thank you for your comments and feedback!
>
> The initial title of the paper might have been slightly misleading in this instance. We are not embedding states and actions into the same embedding space. Instead, we have two separate embedding spaces, one for actions and one for states, but they are jointly learned. We have changed the title of the paper to “Jointly-Learned State-Action Embeddings for Efficient Reinforcement Learning” to avoid this misunderstanding.
>
> **Regarding ” it is unclear if optimizing $\pi_o$ according to Alg 1 can really lead to the optimal one”:** Theorem 1 shows the existence of an optimal policy that can be represented using the internal policy $\pi_i$ and the embedding functions $\phi$ and $f$. We then proceed to show that this can be learned by directly updating the internal policy $\pi_i$ in Lemma 2. The logic behind the proof for this is as follows: Using our Assumptions 1 and 3, we can show equivalence between the internal state-action-value function, using the embedded state and action, and the overall state-action-value function, using the state and action in the original problem domain. Using this equivalence, we can remove the state embedding function $\phi$ and the action mapping function $f$ from the computation of the gradients for the overall policy and are then left with the gradients of the internal policy $\pi_i$ only. This shows that an overall policy can indeed be learned by updating the internal policy directly as per Algorithm 1. One of the contributions of our work is that we show that working directly in the embedding space and updating the internal policy works, rather than using some embedded representations without this theoretical result.
>
> **Regarding “Shouldn’t it be the case that one embedding can correspond to multiple actions?”:** The key idea behind our embedding approach is that states and actions are embedded into a continuous space, where their embeddings are determined by the structure of the environment learned by the embedding model. The agent takes the embedded state as input and outputs a point in embedding space. This point in embedding space is then mapped to an action in the original problem domain. One could think of this as areas in action embedding space mapping to an action in the original problem domain. If we consider environments with discrete actions, then generalization is enabled by mapping the original actions into a continuous embedding space, where they are structured according to their effect in the environment. We are therefore not summarising multiple actions into a single embedding, but instead structure actions in a continuous action embedding space, where generalization is possible. This generalization is mainly enabled by similar actions (in terms of their effect on the environment) being close in embedding space, which they might not be in the original problem domain. We have added a visualization of the learned embeddings for a gridworld domain in the paper, which might help clarify this.
>
> **Regarding “Whenever taking both state and action as input for the critic/value network, one can think of the final hidden layer as the learned joint state-action representation.”:** The key difference in our approach is that embeddings are learned via a separate model, where we can learn these in a supervised fashion. The approaches you are referring to would learn some internal state action representation via a joint optimization in the RL algorithm rather than learning these in a separate supervised model.
>
> **Regarding “As I mentioned, one baseline is to simply take both state-action as input. Another intuitive baseline is to simply learn state-action representation by using an environment model first and then based on the learned representation to learn a policy.” :** We do not believe that these would be appropriate baselines here, as they assume that states and actions are embedded into the same space, which they are not in our approach. Since our approach can be used with any policy gradient based RL algorithm suitable for continuous control, we evaluate our method against this baseline. Additionally, we implemented the action embedding methodology proposed by Chandak et. al. (2019) for domains with discrete action spaces as an additional baseline. The second baseline you suggest is more or less what we propose. We use an embedding model to learn embeddings for states and actions and then train an RL agent using these embedded representations of states and actions. We hope that we were able to clarify this misunderstanding. We also added an **additional experiment on the continuous domain Ant-v2 from the Mujoco physics engine**, where our approach outperforms the baselines.
>
> (References are in the paper)

---

### Official Review · AnonReviewer1 · 2020-10-28
**Official Blind Review**

**Rating:** 5
**Confidence:** 5

**Review:**

This paper explores the idea of using state and action embeddings for more scalable learning. The idea has merit so I encourage the authors to continue working in this direction, but unfortunately there are a number of technical issues with the paper that I detail below.

The main issues for me are:
1. **There are a lot of relevant references missing.** With regards to planning over latent spaces here are some:
  - "Recurrent World Models Facilitate Policy Evolution", David Ha and Jürgen Schmidhuber, NeurIPS 2018.
  - "Model-Based Reinforcement Learning for Atari", Lukasz Kaiser et al. ICLR 2020
  - "Learning Latent Dynamics for Planning from Pixels", D Hafner, T Lillicrap, I Fischer, R Villegas, D Ha, H Lee, J Davidson, ICML 2019
  - "Mastering Atari with Discrete World Models", D Hafner, T Lillicrap, M Norouzi, J Ba, arXiv 2020

  The authors seem to be learning equivalence relations over the states and actions in MDPs (this is basically the result of Assumption 2). There is a large body of work on this, and here are some important references:
  - Givan, R.; Dean, T.; and Greig, M. 2003. "Equivalence notions and model minimization in Markov decision processes". Artificial I Intelligence 147(1-2): 163–223.
  - Ferns, N.; Panangaden, P.; and Precup, D. 2004. "Metrics for finite Markov decision processes". In Proceedings of the 20th conference on Uncertainty in artificial intelligence, 162–169. AUAI Press.
  - Li, L.; Walsh, T. J.; and Littman, M. L. 2006. "Towards a Unified Theory of State Abstraction for MDPs. In ISAIM.
  - Taylor, J.; Precup, D.; and Panagaden, P. 2009. "Bounding performance loss in approximate MDP homomorphisms". In Advances in Neural Information Processing Systems, 1649–1656.
  - Castro, P. S.; Panangaden, P.; and Precup, D. 2009. "Notions of state equivalence under partial observability". In Proceedings of the 21st International Joint Conference on Artificial Intelligence (IJCAI-09).
  - Castro, P. S. 2020. "Scalable methods for computing state similarity in deterministic Markov Decision Processes". In Proceedings of the AAAI Conference on Artificial Intelligence.
  - Zhang, A.; McAllister, R.; Calandra, R.; Gal, Y.; and Levine, S. 2020. "Learning Invariant Representations for Reinforcement Learning without Reconstruction. arXiv preprint arXiv:2006.10742 .

2. **Lemma 1 is not correct.**
  - Consider the following simple example: an MDP with two states ($s, t$) and two actions ($a,b$). Action $a$ always transitions deterministically to state $t$, while action $b$ always transitions deterministtically to state $s$. In state $s$ taking action $a$ gives a reward of 1 while taking action $b$ gives a reward of 0; in state $t$ taking action $a$ gives a reward of 0 while taking action $b$ gives a reward of 1. The optimal policy is to take action $a$ from state $s$ and action $b$ from state $t$, resulting in $V^*(s) = V^*(t) = \frac{1}{1-\gamma}$. Now, take $X = \lbrace x_1, x_2\rbrace$ where $\phi(s) = x_1$ and $\phi(t) = x_2$ (this satisfies Assumption 2). Now take $E = \lbrace e\rbrace$ where $f(e) = a$, so $f^{-1}(a) = \lbrace e\rbrace$ (this satisfies Assumption 1). Letting $g:A\rightarrow E$ be the action embedding function, we can see that $g(a) = g(b) = e$, which means that any internal policy will be suboptimal, as it will choose the same latent action for both $x_1$ and $x_2$.
  - Part of the issue may lie in the way the proof is structured. The authors start by using $G$ which they define as "denotes the return, which is a function of $s$, $a$, and $s'$". However, the way they've included it in the definition of $v^{\pi}$ seems like it does _not_ depend on $s'$ and is really just $Q^{\pi}(s, a)$. This seems to be the case as that is where the authors end up at the end of the proof. If it _is_ supposed to depend on $s'$, then $G$ likely needs to be decomposed into the one-step reward $R(s, a)$ and the expected value at the next state $\mathbb{E}V^{\pi}(s')$.
  - The other bug I found in the proof is at the top of page 12. The authors went from $P(a | s, a')$ at the end of page 11 to $P(a, x | s, a')$ at the top of page 12, which is most certainly not a valid equality.
  - (Minor) The jump from $P(s, a)$ to $\pi(a | s)P(s)$ is probably ok, but requires a little more justification.

3. **The losses are incorrect**. Below equation (4), the authors state "the denominator does not depend on $\phi$, $g$, or $T$, and so they get rid of the $P(S_{t+1}|S_t, A_t)$ term in their loss, but this means there is no target for the loss! This means that the loss in equation (5) is just trying to maximize the probabilities uniformly, independent of whatever the true probabilities really are. This problem is also present in the loss in equation (6).

Medium issues:
1. The authors make a number of claims about their method working for continuous spaces, but this requires more details than simply replacing integrals for sums. For example, in the background the transition function T needs to be defined with respect to Borel sets, the summation in Lemma 1 doesn't "just work" by switching to integrals, etc..
2.  Given that the authors motivate their work by claiming that existing algorithms don't work well outside of "simple" tasks, they should include larger scale experiments than the ones they are currently including, which are rather small.
3. Doesn't Assumption 3 defeat the purpose of dealing with large state spaces? It's basically just converting the original state space into an isomorphic one.

Minor issues:
1. In the Introduction the authors refer to the ALE as "comparatively simple tasks", but they are not simple and are in fact more difficult than any of the environments evaluated in the paper.
2. In equation (3) it should be $\hat{T}(X_{t+1} | X_t, E_t)$ instead of $\hat{T}(S_{t+1} | X_t, E_t)$

---

> ### Author Response · Authors · 2020-11-24
> **Response: Main Issue 1**
>
> Thank you for your comments and feedback!
>
> ## Main issue 1
>
> Thank you for pointing these references out! The two strands of research you point out are indeed relevant. Both world models (used to generate state representations or to evaluate a policy using the learned world model), as well as the research related to bisimulation-based state aggregation, are highly related to our work. Apparently, we missed this in our literature research. We have tried to rectify this issue by including a discussion of the branches of research you reference in the introduction and in the related work section of our revised paper.
> Compared to the existing work on world models, our approach differs in the following ways: Firstly, we learn embedded representations for both states and actions, rather than just for states as done in the world model literature. Secondly, we do not use the model of the environment to do policy evaluation, but instead, learn directly in the original environment, but using an abstract (embedded) representation of the environment.
> Compared to the bisimulation approaches to state aggregation, our work differs in the following ways: Firstly, we learn representations (embeddings) for both states and actions rather than for states alone. Secondly, we do not explicitly aggregate states, i.e., we do not treat two states as exactly the same, but instead project them into a continuous embedding space, where we capture the similarity between states. In this sense, our approach has some similarities to the bisimulation methods you reference.

---

> > ### Author Response · Authors · 2020-11-24
> > **Response: Main Issue 2**
> >
> > ## Main Issue 2
> >
> > - Your second point seems to be based on a misunderstanding.
> > The action embedding function $g$ is a one-to-one mapping from actions in the original problem domain to action embeddings.
> > The action mapping function $f$ is a many-to-one mapping from action embedding space to actions in the original problem domain. One can think of this as areas in action embedding space mapping to a certain action in the original problem domain.
> > The inverse of this, i.e., $f^{-1}$, is then defined as $f^{-1}(a) := \set{e \in \mathcal{E} : f(e) = a}$ and is therefore a one-to-many mapping.
> > Moreover, the state embedding function $\phi$ is a one-to-one mapping from states in the original problem domain to state embeddings.
> > Following your example, we have a state embedding for each of the states in the original problem domain: $X = \set{ x_1, x_2 }$ (satisfies Assumption 3).
> > In your example, however, Assumption 1 would not be satisfied. If we had a single embedding $E = \set{e}$ that corresponded to both actions $a$ and $b$, i.e., $g(a) = g(b) = e$, then we would not be able to map the point $e$ in embedding space deterministically with $P(A_t = a | E_t = e) = 1$ since there would now be two candidate actions $A_t$ for $E_t = e$, thus **not** satisfying Assumption 1. The situation you describe, i.e., two actions being mapped to the same embedding thus cannot arise as per Assumption 1. For your reference, the argument we make here using Assumption 1, is the same as in Chandak et. al. 2019 at the end of Lemma 2. They use the same Assumption (Assumption 1) to derive a similar result on the use of action embeddings. To clarify this further, we have tested that Assumption 1 holds in practice, by checking that no two actions share exactly the same embedding in the experiments we conduct. We have added this to the revised paper.
> >
> > - The definition of return $G$ is equivalent to what you suggest. Return is commonly defined as $G = \mathcal{R}(s, a) + \gamma v^{\pi}(s')$. The Q function is commonly defined as $Q^{\pi} (s, a) = E_{\pi}[G_t | S_t = s, A_t = a]$. The Q function therefore still depends on $s, a, s’$. Since we know $s$ and $a$, the Q function can then be written as $\sum_{s’ \in \mathcal{S}} P(s’ | s, a) [\mathcal{R}(s, a) + \gamma v^{\pi}(s')]$, which is equivalent to $\sum_{s’ \in \mathcal{S}} P(s’ | s, a) G$. This is exactly what we do in the paper. In response to your comment, we have included our definition of $G$ in the proof of Lemma 1 in the appendix to clarify this.
> >
> > - This is not a bug. Since we do not use $a’$ in our proof of Lemma 1, we assume that you are referring to the step from $P(a | s, s’)$ to $P(a, x | s, s’)$. For the conditional probability $P(a|s,s’)$, we have $P(a|s,s’) = P(a,s,s’)/P(s,s’)$. Then by Assumption 2, $\phi(s)$ is a deterministic mapping given $s$. We therefore have $P(a,s,s’) = P(a,s,s’,\phi(s))$, since $P(\phi(s) | s) = 1$ by Assumption 2. Using the notation in this paper, $P(a,s,s’,\phi(s))$ is the same as $P(a,s,s’,x)$. Therefore, $P(a|s,s’) = P(a,s,s’)/P(s,s’)=P(a,s,s’,x)/P(s,s’)=P(a,x|s,s’)$. Hence, our equation is correct.
> >
> > - This conversion is also correct, but we understand that the notation might be slightly confusing here. $\pi(a | s)$ is essentially a distribution over actions given states, i.e., it is still a probability distribution, just using the policy notation. Therefore, $\pi(a | s) P(s) = P(a|s)P(s) = P(a, s)$. We chose to use $\pi( a| s)$ instead of $P(a|s)$ to make the fact that this is our policy explicit to the reader.

---

> > > ### Author Response · Authors · 2020-11-24
> > > **Response: Main Issue 3**
> > >
> > > ## Main Issue 3
> > >
> > > The loss we use is a fairly standard negative log-likelihood loss function. The derivation from the KL Divergence is as follows:
> > > $$
> > > D_{KL} (P || \hat{P}) = - E_{S_{t+1}\sim P(S_{t+1}| S_t, A_t) } \Big [\ln \Big ( \frac{\hat{P}(S_{t+1} | S_t, A_t)}{P(S_{t+1} | S_t, A_t)} \Big ) \Big ]
> > > $$
> > > $$
> > > = - E_{S_{t+1}\sim P(S_{t+1}| S_t, A_t) } \Big [\ln \hat{P}(S_{t+1} | S_t, A_t) - \ln P(S_{t+1} | S_t, A_t) \Big ]
> > > $$
> > > $$
> > > = - E_{S_{t+1}\sim P(S_{t+1}| S_t, A_t) } \Big [\ln \hat{P}(S_{t+1} | S_t, A_t) \Big ] + E_{S_{t+1}\sim P(S_{t+1}| S_t, A_t) } \Big [ \ln P(S_{t+1} | S_t, A_t) \Big ]
> > > $$
> > > Here, term $E_{S_{t+1}\sim P(S_{t+1}| S_t, A_t) } \Big [ \ln P(S_{t+1} | S_t, A_t) \Big ]$ does not depend on any of our model parameters, so we can drop it. Then the remaining term
> > > $- E_{S_{t+1}\sim P(S_{t+1}| S_t, A_t) } \Big [\ln \hat{P}(S_{t+1} | S_t, A_t) \Big ]$ corresponds to the negative log likelihood. Therefore, the target we define in our paper corresponds to maximum likelihood estimation (MLE). In response to your question, the ground truth information is included in the subscript of the expectation. Sometimes, this target is explicitly conditioned on the model parameters, e.g. something like $- E_{S_{t+1}\sim P(S_{t+1}| S_t, A_t) } \Big [\ln \hat{P}(S_{t+1} | S_t, A_t, \Theta) \Big ]$, where $\Theta$ are the model parameters. We omit this explicit conditioning to simplify the expression.
> > > In response to your comment, we have tried to clarify this further in Section 4.3.1 and the subscript of the expectation is added to avoid confusion.

---

> > > > ### Author Response · Authors · 2020-11-24
> > > > **Response: Medium and Minor Issues**
> > > >
> > > > ## Medium Issues
> > > >
> > > > 1. The internal policy $\pi_i$ is already defined over continuous embedding space, i.e., it already is a probability density function (PDF).
> > > > The definition of the Q function also does not change for continuous vs. discrete state and action spaces. In both cases, it is defined as an expectation of rewards over next states, rewards, and actions, given the current policy.
> > > > Consequently, these two components do not need to be defined differently when changing from discrete to continuous action spaces.
> > > > The action mapping function $f$ also does not depend on whether the action space is discrete or continuous. The only requirement we have for $f$ is given by Assumption 1, i.e., it deterministically maps each point in embedding space to an action (or a point in the action space of the original problem domain for continuous action spaces) with $P(A_t = a | E_t = e) = 1$.
> > > > Hence, there are no special restrictions on the embedding space $\mathcal{E}$ that would necessitate a different definition for continuous action spaces.
> > > > One implicit change when replacing $\sum_{s \in \mathcal{S}}$ and $\sum_{a \in \mathcal{A}}$ by $\int_s$ and $\int_a$ is that the probability mass functions (PMF) in Lemma 1 and Lemma 2 would now become PDFs for the continuous domain.
> > > > Therefore, the theoretical results obtained for discrete states and actions still hold for continuous domains by making this simple change.
> > > >
> > > > 2. We divide our experiments into two types, namely proof-of-concept (toy) domains, such as the gridworld and slotmachine domains, and larger domains such as the recommender system. The former primarily serve as an illustration of how the approach works, while the latter highlight the method’s applicability to real-world problems. We believe that the experiments on the real-world recommender system (which is similar to that used in Chandak et. al. 2019) demonstrate the efficiency of our approach in real-world-scale settings with large discrete state and action spaces. In response to the reviews, we have now included a further experiment on the continuous Ant-v2 environment from the Mujoco physics engine, which is significantly more complex than the previously included continuous half-cheetah environment.
> > > >
> > > > 3. Assumption 3 states that no two states share exactly the same embedding. However, generalization in the state embedding space in our approach is not achieved by aggregating similar states into the same state (as for instance in some of the bisimulation-based approaches), **but by capturing similarity by mapping similar states into similar regions of the state embedding space.** By embedding states (and actions), our approach structures these according to the characteristics of the environment. This structure is usually not given in the original state and action representation - especially in discrete domains. In that sense, it is not an isomorphism. The key factor enabling generalization in the embedding spaces is therefore that similarity between states and actions is captured.
> > > >
> > > > ## Minor Issues
> > > >
> > > > 1. Thanks for pointing this out! We have rephrased this in the revised paper. The main goal of mentioning this was to demonstrate the rare application of SOTA reinforcement learning methods to real-world domains, rather than depicting ALE domains as simple tasks.
> > > >
> > > > 2. The original expression is correct in this case. The transition model does not predict states in state embedding space but in the original problem domain.
> > > >
> > > > (References are in the paper)

---

> > > > > ### Comment · AnonReviewer1 · 2020-11-24
> > > > > **Medium and minor issues response**
> > > > >
> > > > > Thanks for your responses. Some further comments below.
> > > > >
> > > > > 1. You say "the probability mass functions (PMF) in Lemma 1 and Lemma 2 would now become PDFs for the continuous domain. Therefore, the theoretical results obtained for discrete states and actions still hold for continuous domains by making this simple change.", which is the point I was originally raising. In order to be a proper measure you need to define a proper probability space (with something like Borel sets). The choice of this affects how well your results carry through. It's likely they do just carry through, but it is not currently presented in a mathematically rigurous fashion.
> > > > > 2. I appreciate that Ant-v2 is more difficult than half-cheetah and thanks for including that extra experiment. However, these are all still rather toy-ish domains, so I would perhaps soften the claims about other methods not working well with real-world tasks: the empirical evidence you're currently providing doesn't quite hit the mark in terms of showing that your method _does_ work in real-world settings.
> > > > > 3. See my point above. I agree that one would expect that learning structured embedding spaces nearby points are similar should lead to better generalization, but this is not what your theory is showing. As mentioned previously, your theory is dealing with isomorphisms; whether or not points are near to each other is not a component considered in your theoretical results. As such, it is not possible to make claims about better generalization other than through empirical evidence.
> > > > > Incidentally, this is why previous works use metrics, as the distance between states can be directly leveraged to infer claims of generalizability.

---

> > > > > > ### Author Response · Authors · 2020-11-24
> > > > > > **Response to Medium and Minor Issues Comments**
> > > > > >
> > > > > > Thanks for your further comments!
> > > > > >
> > > > > > 1. What we described is the high-level method on how to apply our theoretical results to the continuous domain. Due to the time and page constraints, we did not detail each mathematical step on the continuous RL problem in the paper. Nevertheless, we can certainly write down all these mathematical details in the appendix in a future version.
> > > > > >
> > > > > > 2. In the revised paper, we are more careful about claiming that our approach outperforms on real-world domains. However, we motivate our approach from the need to develop methods to deal with large state/action spaces to foster the applicability of RL to real-world problems. Furthermore, we consider the recommender system, with $835$ actions and approx. $700,000$ discrete states a real-world problem, where our approach outperforms the baselines.
> > > > > >
> > > > > > 3. Yes, as you point out, the theory is not about proving the distance of nearby points in the embedding spaces. Regarding the distance of points in the embedding spaces, we only have empirical studies, such as the visualization in Fig. 2. However, the ultimate goal of our theoretical result here is to prove that by our embedding method, we can purely focus on the gradient-based update of the internal policy $\pi_i$ (Lemma 2), and such an update of the internal policy is theoretically correct as it leads to an overall policy $\pi_o$ that achieves optimality in the original problem domain (Theorem 1). Note that at the end of the day, we want to give a policy that will work in the original problem domain, and our theoretical results show that we can achieve that by our method.

---

> > > > > > > ### Comment · AnonReviewer1 · 2020-11-24
> > > > > > > **Generalization versus optimality**
> > > > > > >
> > > > > > > I agree with your comments regarding point 3, but my point remains: your theory shows that you can achieve optimality but does not say anything about generalization. This is evaluated in the empirical section, but I'm focusing on the theoretical claims.
> > > > > > >
> > > > > > > I think there's a tension between optimality and generalization here.
> > > > > > >
> > > > > > > **Optimality**
> > > > > > > You say "at the end of the day, we want to give a policy that will work in the original problem domain", but isn't the main motivation that you want a method that is able to generalize well in very large systems?
> > > > > > >
> > > > > > > **Generalizability**
> > > > > > > In contrast, you also state "the ultimate goal of our theoretical result here is to prove that by our embedding method, we can purely focus on the gradient-based update of the internal policy $\pi_i$ (Lemma 2)", but Lemma 2 requires $\phi$ to be a one-to-one mapping, which is in conflict with the notion of generalization.

---

> > > > > > > > ### Author Response · Authors · 2020-11-24
> > > > > > > > **Response: Generalization vs Optimality**
> > > > > > > >
> > > > > > > > Thanks for your comment.
> > > > > > > >
> > > > > > > > **Generalization vs Optimality**
> > > > > > > >
> > > > > > > > Our theoretical results show optimality, for which we require Assumptions 1 and 3. However, this is not in conflict with improved generalization, since Assumptions 1 and 3 do not prevent the method from capturing similarities in embedding space. You are, however, right in that we do not show better generalization theoretically. This is only investigated empirically.
> > > > > > > > Given the current time constraint, experiments on very large systems will be considered for future work.
> > > > > > > >
> > > > > > > > Another note on **Generalization**:
> > > > > > > > In the experiments we conduct, Assumption 3 holds. However, we can not be absolutely certain that this will also be the case for other environments.  Meanwhile, we want to point out that even if assumption 3 is violated a little bit, as long as the two gradients in Lemma 2 are similar, we believe Algorithm 1 can still be applied with reasonably good performance; the detailed validation will be left for future work.

---

> > > > ### Comment · AnonReviewer1 · 2020-11-24
> > > > **Thanks for clarifying**
> > > >
> > > > Thanks for clarifying, I think this was related to my minor point, where I thought your model was predicting states in the embedding space instead of the original domain.

---

> > > ### Comment · AnonReviewer1 · 2020-11-24
> > > **Thanks for clarifications**
> > >
> > > Thanks for clarifying, this does clear up my doubts about the theoretical correctness of your results so I no longer think they are incorrect.
> > >
> > > My doubts regarding the utility of these results still remain, however, given that the one-to-one assumptions _do_ result in an isomorphic MDP. Below you state that they are "structured" in that similar states will be nearby in the embedding space. This may be the case and presumably leads to better generalization, but your theoretical results are not demonstrating that. In particular, the correctness and Lemma 1 and Theorem 1 only work with the one-to-one assumption (e.g. no generalization).
> > >
> > > I can appreciate that theoretical results on generalization with deep nets is difficult, but at that point the evaluation of your method is mostly empirical. Given that the domains considered are small- to mid-scale, and the fact that you are not comparing against related work such as Zhang et al. (2020), the empirical evaluations are not as strong as they could be.

---

> > > > ### Author Response · Authors · 2020-11-24
> > > > **Response to comment**
> > > >
> > > > According to Assumption 1, $g$ is a one-to-one mapping.
> > > > However, $f$ is still a many-to-one mapping, as $\pi_i$ will generate action embeddings in a continuous space. Moreover, for the correctness of Lemma 1 and Theorem 1, we only need $\phi$ and $f$ to be many-to-one mappings. Assumption 3 (which defines $\phi$ as a one-to-one mapping) is only needed for the gradient result in Lemma 2.
> > > > Your pointer to the work by Zhang et. al. (2020) is much appreciated. Due to the limited time, we cannot compare our approach to their method, but this will be considered for future work.

---

> > ### Comment · AnonReviewer1 · 2020-11-24
> > **Added references**
> >
> > Thank you for adding the references, it certainly helps position the paper better.
> > However, I take issue with a statement you make in the Related Work section. After discussing (Zhang et al., 2020) and (Castro, 2020), you say "While there are parallels between bisimulation and our approach, we do not propose the aggregation of statess. Instead, our embedding technique projects states into a continuous state embedding space, where their behavioral similarity is captured in their proximity in embedding space." But this is _exactly_ what (Zhang et al., 2020) did! The difference that I do agree with is that they only embed states, not state-actions.
> >
> > Given how late it is in the game, it's unreasonable for me to ask for you to compare against Zhang et al.'s method. Nevertheless, that is a clear point of comparison that would make the results more convincing.

---

> > > ### Author Response · Authors · 2020-11-24
> > > **Rephrased**
> > >
> > > Thanks for pointing this out. This sentence was not intended specifically in relation to Zhang et. al. 2020, but to some other works using bisimulation metrics that we mention. We have rephrased this to "Instead, our embedding technique projects states into a continuous state embedding space, similar to Zhang et al. (2020), where their behavioral similarity is captured in their proximity in embedding space."

---

> ### Comment · Area_Chair1 · 2020-11-24
> **Please engage in the conversation**
>
> Dear reviewer,
>
> Please let us know whether the authors' rebuttal to your questions are satisfactory or not. If you need more clarifications on any issue, please ask your questions as soon as possible. Today (Nov. 24) is the last day that the authors can reply back to you.
>
> Thank you,
> AC

---

### Official Review · AnonReviewer4 · 2020-10-29

**Rating:** 4
**Confidence:** 4

**Review:**

The paper proposes a framework of jointly learning a state and action embedding using the model of the environment, eventually using those embeddings to learn a parameterized control policy using standard policy gradient (PG) methods. Joint learning of state and action embeddings allows us to capture the interactions between actions in different states. The framework proposes to learn an internal (embedding) policy, a state embedding, an inverse function on action embeddings, combining all the parts to form an overall policy. The paper theoretically shows that optimizing the internal policy leads to an optimal overall policy.

The idea presented in the paper is a natural extension to work done in Chandak et al. (2019). The idea of jointly optimizing for the state and action embedding, as opposed to just separately optimizing the action embedding as in Chandak, is reasonable and warrants investigation. As is, however, the paper needs a bit more work for three primary reasons. Firstly, the embedding for states and actions involves a joint optimization, but the embedding itself seems to be separate for the two. The relationship between how the state embeddings influence the action embedding is unclear. The inverse embedding function ‘f’ does not seem to account for the state when inverting the embeddings, and the f seems to be a global unembedding (decoder) across all states. The properties of the learned embedding could be better explained.

Second, the experiments do not clearly highlight why the joint state and action embedding might help learn a good policy. For example, Chandak et al. (2019) present experiments, where they visualize the benefit of using action embeddings as part of the internal policy. Including experiments which show meaningful relationships between states and their corresponding action embedding will add soundness to the benefits of the proposed framework.

Finally, the empirical results are inconclusive about the benefits of JSA. Five random seeds are not adequate to draw relevant conclusions given there is extensive overlapping between standard errors.  As pointed out in Henderson et al. (2017), random seeds vastly influence the performance of methods.

Comments:
1. Page 1, Para 3 mentions the proposed method to bridge the gap between model and model-free learning. This needs clarification, as in model-based approaches, the model is employed for planning, whereas, in this case, it's used to learn embeddings.

2. Section 2 (background) : The distributions in this work seem to be defined as functions to the set, rather than as a distribution. For example, the transition function is defined as probability of transition to the next state from current state and action, but the terminology defines it as a deterministic function i.e. \mathcal{T} : \mathcal{S} \times \mathcal{A} \rightarrow \mathcal{S}. Either define it as deterministic or use \mathcal{T} : \mathcal{S} \times \mathcal{A} \times \mathcal{S} \rightarrow [0,1]. In Equation (3), these are then used as distributions, rather than deterministic functions.

3. If the space permits, it is good to include an example where a joint state-action representation will help in learning the policy.

4. In Theorem 1, the optimal value function is defined as a function of \pi_i and Q^*. It would be more clear to use separate notation to represent the optimal set of internal policies, maybe \pi_i^*.

5. Looking at Figure 2(a,b,d) JSA seems to have worse asymptotic performance. This could be discussed.

6. The following missing citation looks relevant: “Reinforcement Learning with Function-Valued Action Spaces for Partial Differential Equation Control”, Pan et al., 2018. (see reference 3 below)

7. It's good to see many of the experimental details mentioned. I would recommend including the following details:
	a. Size of batch and mini-batch used.
	b. Epsilon value, for importance sample clipping in PPO.

Questions:
1. Algorithm 1 mentions online training of the transition model, but the Appendix notes the use of pretraining the model for embeddings. Which one is used?

2. In the case of the recommender system experiments, it is unclear to me how it is an RL problem i.e. it seems that the agent is defining the probability transition function, which would be wrong. The setting seems more relevant to pose as a contextual bandit. Also, more details on the reward function would be helpful; it seems like the agent can earn a high reward, even for wrong predictions, given the user is purchasing expensive items.

3. Gridworld experiments: There are some details which are missing from the environment. For example, what is the exact reward function? Looking at the plots from Chandak et al. (2019), the reward scaling is near 100, and in the current paper figures for Gridworld rewards are near 1.

4. When you say you train agent for 700 epochs with 1500 steps, do the steps refer to gradient descent steps and does 700 epoch mean that there 700 * 10 = 7000 episodes at a minimum? I was unsure about the connection between epochs and the number of episodes.

5. How are the state and actions fed into the transition model (e.g., are they concatenated, etc.)?

References
1. Chandak, Y., Theocharous, G., Kostas, J., Jordan, S., & Thomas, P. S. (2019). Learning Action Representations for Reinforcement Learning. http://arxiv.org/abs/1902.00183

2. Henderson, P., Islam, R., Bachman, P., Pineau, J., Precup, D., & Meger, D. (2019). Deep Reinforcement Learning that Matters. http://arxiv.org/abs/1709.06560

3. Pan, Y., Farahmand, A., White, M., Nabi, S., Grover, P., & Nikovski, D. (2018). Reinforcement Learning with Function-Valued Action Spaces for Partial Differential Equation Control. http://arxiv.org/abs/1806.06931

----- Update

Thank you for the update and response. Unfortunately, some of my concerns remain. The plots are now run with 10 runs, rather than only 5 runs in Figure 5. But, they look almost identical (in some cases, maybe they are identical?). That is not possible, unless there is a potentially invalid choice in the experimental design. If nothing else, the standard error should change.

The theory itself has some utility, since it is shown that learning in embedding space is equivalent. This is not surprising, considering it is assumed that there is a one-to-one mappings, but it’s good to be thorough. Nonetheless, this could maybe be shown more simply, and I am not sure Lemma 2 is exactly correct.

Lemma 1 is overly complex.

Alternative proof:

Assume pr(a | s) is pi(a | s) (i.e., action probabilities given s are defined under pi).

vpi(s) = sum_a pr(a | s) qpi(s,a) = sum_a int_e pr(a, e| s) de qpi(s,a) = sum_a int_e pr(a | e, s) pr(e | s) de qpi(s,a) = sum_a int_e pr(a | e) pr(e | s) de qpi(s,a) (also using Claim 5 like they do) = sum_a int_{f^{-1}(a)} pr(e | s) de qpi(s,a) = sum_a int_{f^{-1}(a)} pr(e | phi(s)) de qpi(s,a)

Lemma 2 claims to show that the gradients are equivalent, but instead it seems to show that the functions themselves are equivalent and so should maybe be stated that way. Gradients are just placed in front of everything. Further the last step replacing d_0(s) with d_0(x) seems incorrect, as s is from a discrete space and x from a continuous space. Maybe you are suggesting that d_0 is some kind of delta distribution, but then it might be better to just sum over the same set of s.

I am also a bit unsure about any smoothness assumptions required. Is J_0 even differentiable in theta? The requirements on the one-to-one mappings between discrete state to continuous state make for a piecewise flat function that could be problematic for such gradients.

I also appreciate that Figure 2 was added. But, it is a bit hard to interpret. More explanation is needed there.

---

> ### Author Response · Authors · 2020-11-24
> **Response: Comments**
>
> Thank you for your feedback and comments!
>
> We have tried to address your points in the paper, please see below for the concrete changes and additions that were made.
> We have changed the title to “Jointly-Learned State-Action Embeddings for Efficient Reinforcement Learning” to avoid any misunderstandings.
> We have increased the number of random seed runs to 10 random seeds. From other related research, e.g. Chandak et. al. 2019, this seems to be a common number of random seed runs.
> We have also added a visualization of the learned embeddings for states and actions, similar to that in Chandak et. al. 2019, which illustrates how the learned embeddings capture the underlying structure in the state and action space.
>
> ## Comments
>
> 1. Indeed, we do not use an environment model for planning. The notion of “bridging the gap” stems from the fact that we learn a model of the environment to generate embeddings. We have clarified this distinction in the revised paper (page 1, para3).
>
> 2. Thanks for pointing this out! We adjusted the definition of the transition function slightly to avoid this misunderstanding. Rather than defining a probability, the transition function - in the way we define it - defines the transition from one state to another given an action.
>
> 3. Hopefully, comment 3 has been addressed by visualizing the learned state and action embeddings in the revised paper (new Fig. 2).
>
> 4. We believe that this would be slightly incorrect here. In Theorem 1, we are concerned with an optimal policy in the original problem domain. The optimal state-action-value function $Q^*$ in Theorem 1 is in the original problem domain, i.e., not in embedding space. Overall, we are still **concerned with finding an optimal policy in the original problem domain**. Theorem 1 shows that this can be done by optimizing the overall policy $\pi_o$, which consists of the internal policy $\pi_i$ and the embedding model.
>
> 5. We conjecture that the comparable or worse asymptotic performance in continuous problem domains might be due to the structure that is already given in continuous representations, i.e., some ordering/meaning being inherent to continuous actions and states. This would make it harder for the embedding model to discover additional structure beyond what is already present in the state and action representation. We have added some more details on this in the paper. In addition, we also ran an **additional experiment on the Ant-v2 environment** from the Mujoco physics engine, where our approach outperforms the benchmarks (see Fig. 3 in the revised paper).
>
> 6. Thanks for the suggestion! We did have a look at the paper you mentioned and it seems that this work is concerned with enabling RL for function-valued action spaces rather than with learning state or action representations. We therefore do not think that this is particularly relevant to our approach.
>
> 7. Thanks for pointing this out! We have included more details, including the best set of hyper parameters for each experiment, in the appendix.

---

> > ### Author Response · Authors · 2020-11-24
> > **Response: Questions**
> >
> > ## Questions
> >
> > 1. For the proof-of-concept experiments, we only pre-train the embeddings and keep them fixed thereafter. This includes the gridworld, slotmachine, and half-cheetah. Since these environments are relatively easy to tackle, we decided to go with minimal experiments here. For the more complex environments, i.e., the recommender system and the new Ant-v2 environment, continuous updates are enabled in addition to pre-training. We have clarified this in the updated appendix.
> >
> > 2. The recommender system environment is implemented according to Shani et. al. (2005), who propose this as a reinforcement learning problem. In the environment model, we compute the purchasing probability for a user for each item. The RL agent then recommends an item. This scales the probability of that user purchasing that item by a pre-defined factor. However, the initial probabilities depend on the user. Therefore, the agent does not define the transition probability but impacts it. We chose this environment as it presents a problem with large discrete state and action spaces, which is well suited for our approach. Additionally, Chandak et. al. (2019) also use a recommender system following Shani et. al. (2005) for their experiments and we hope that this fosters comparability.
> >
> > 3. We have added these details to the appendix.
> >
> > 4. Following the reference implementation from OpenAI’s Spinninup package, training is conducted in epochs rather than updating the models after every timestep or every episode. We have clarified this in the appendix.
> >
> > 5. The states and actions are first fed into the respective embedding models. The embedded representations are then concatenated before being fed into the transition model. We have added this detail in Section 4.3.1.
> >
> > (References are in the paper)

---

### Official Review · AnonReviewer5 · 2020-11-06
**possibly has merit, but limited impact with the current evaluations**

**Rating:** 3
**Confidence:** 3

**Review:**

### Summary
The paper proposes a method to jointly learn: (a) a latent state embedding; (b) a latent action embedding; (c) a state transition model; and (d) an RL policy.  The latent models should allow for better generalization over states and actions, and therefore result in improved learning, particularly for discrete action domains. The method shows improved performance over vanilla policy gradient on a grid-world task, a slot machine task, a recommender system, and half-cheetah locomotion.

### Strong points
- important problem: learned latent models of the world are key to making progress in RL
- evaluated on several tasks, with improvements on some
- it is surprising that the all of the components can be trained, in a stable fashion, without further regularization or conditioning

### Weak points
- there is a large and growing literature on leveraging learned latent models of all kinds, particularly for continuous systems. E.g., world models and many variants. It is unclear to this reader how to situate this work in relation to those.
- the evaluation is weak:  two of the main examples are toy proof-of-concept; half-cheetah shows no benefit; the recommender system needs to be considered in the expansive volume of recommender system algorithms
- the method is only compared against vanilla policy gradient for a number of the results

### Recommendations
Currently recommend to reject.  The approach needs to be discussed and evaluated in the context of other latent "world models", and to show benefits on more challenging problems. I also currently remain unclear regarding the potentially underdetermined nature of learning all the given components in the absence of further regularization or constraints.

### Questions
Q1. 4.3.1 component (iii): function g:  How is g well-posed, given that it is inverting a possibly many-to-one mapping?

Q2. Eqn (5):  Given that this is the only loss function that involves three of the models, it is unclear to me how they are fully determined in the absence of further regularization, or additional loss terms.

Q3. Algorithm 1:  Where are the models for g and T updated?

Q4: uniqueness of state embedding:  Why would this emerge as a property? How has this been tested emperically? Doesn't this also mean that some of the benefits of the latent space are lost, if working from possibly-redundant state observations?

Q5:  Figure 2e has comparisons with PPO and SAC. How would these algorithms fare for the other problems?

### Additional feedback

This reader found the phrase "joint action-state embedding" to be ambiguous.  Upon first reading, I interpreted it as learning a unique embedding for the state-action space, e.g., for Q(s,a) for example. Instead, it refers to separate state and action embeddings, which are jointly trained during learning, along with the policy. "Jointly-trained action and state embeddings" would clarify this ambiguity.

Figure 1b: label the transition model, T

4.2 Assumption 1: Given an _action_ embedding

Concluding sentence for the paper:  this makes a very strong statement that is not really supported by the results.

---

> ### Author Response · Authors · 2020-11-24
> **Response: Strong and Weak Points**
>
> Thank you for your comments and feedback!
> ## Strong Points
> Regarding your comment on convergence: We do not use any explicit regularization (e.g. drop-out) in the loss functions or in the network architecture. The implementations for the VPG, PPO, and SAC algorithms we use are based on the reference implementations from OpenAI Spinninup (Achim, 2018). These implementations use a buffer to store experiences collected during an epoch, i.e., multiple episodes and then update the actor and critic network using this buffer (called replay buffer). While this could be considered a kind of regularization, we do not add any further regularization beyond this. For some theoretical results on the convergence of a similar approach, you may find the three-timescale convergence proof in Chandak et. al. (2019) interesting. Another factor that stabilizes the training of the model is the pre-training of the embedding model. In our experiments, we pre-train all components of the embedding model and the action mapping function $f$ using randomly generated trajectories (in Fig. 1-b), before beginning to train the policy network (in Fig. 1-a). Details of this can be found in the Appendix.
>
> ## Weak Points
> 1. Thank you for pointing this out. We have added a review of previous work related to world models and state aggregation techniques using bisimulation metrics to Section 3 (Related Work), where we position our work in relation to these areas.
>
> 2. In the experiments on the recommender system environment, we do not aim to show performance better than that of other specific recommender system algorithms. We chose this domain as it offers large discrete state and action spaces and is thus well-suited to evaluate our approach. Specifically, there are 835 discrete actions and almost 700,000 discrete states in this environment, rendering it a fairly realistic real-world experiment.
> We benchmark our method against several state-of-the-art reinforcement learning algorithms and against the embedding methodology proposed in Chandak et. al. (2019) since these are the algorithms we want to improve over. While it may be interesting to evaluate our approach against other recommender system algorithms, this is not the focus of this work.
> In addition to this, we have now included a **new experiment on the Ant-v2 environment from the Mujoco physics engine**, which is substantially more complex than the previously included half-cheetah environment.
>
> 3. We divide our evaluation into toy and larger/real-world experiments. For simplicity, we only evaluate our approach using vanilla policy gradient (VPG) in the toy domains. For the real-world recommender system, we consider a broader range of RL algorithms, i.e., proximal policy optimization (PPO) and soft actor critic (SAC). Given that the considered toy domains are fairly easy and predominantly serve as a proof-of-concept, we chose to keep the experiments simple by only using VPG. In response to this and other reviews, we have included a further experiment using VPG, PPO, and SAC on the Ant-v2 environment from the Mujoco physics engine, and shown to have superior performance.
>
> (References are in the paper)

---

> > ### Author Response · Authors · 2020-11-24
> > **Response: Questions and Additional Feedback**
> >
> > ## Questions
> >
> > 1. The action embedding function $g$ is a one-to-one mapping from actions in the original problem domain to action embeddings.
> > The action mapping function $f$ is a many-to-one mapping from action embedding space to actions in the original problem domain. This can be thought of as areas in action embedding space being mapped to a certain action in the original problem domain.
> > Following Assumption 1, we require action embeddings (as learned by $g$) to be unique. Consider the following example: If we have two actions $a$ and $b$ with the same embedded representation, i.e., $g(a) = g(b) = e$, then we would not be able to map the point $e$ in embedding space deterministically with $P(A_t = a | E_t = e) = 1$ since there would now be two candidate actions $A_t$ for $E_t = e$, thus not satisfying Assumption 1. Note that the function $g$ should not be thought of as the inverse of the action mapping function $f$. Instead, $g$ is used to obtain embeddings for each action (one-to-one), which are then used to learn/define function $f$, which maps areas in embedding space to actions in the original problem domain.
> > To clarify this further, we have added a visualisation of the learned embeddings in the paper (Figure 2) and conducted an empirical test that Assumption 1 actually holds in practice by verifying that no two action embeddings are exactly the same for all the experiments we ran. This was also added to the paper.
> >
> > 2. In total, we use three loss functions. Firstly, Eq. (5) is used to update the embedding model (model of the environment), i.e., $g$, $\phi$, and $T$. Secondly, the action mapping function $f$ is trained separately, using the loss function in Eq. (6). This loss can be seen as a reconstruction error between the original action $A_t$ and the action reconstructed from the action embedding $E_t$. Thirdly, the policy network is trained by reinforcement learning, using a loss function specified by the respective algorithm used. Therefore, we are able to iteratively train all components of our model using these three loss functions. **It should be noted that they are not jointly trained, but updated in an iterative fashion**, as specified in Algorithm 1.
> >
> > 3. Thank you for pointing this out! The components of the embedding model are iteratively updated in the main loop of Algorithm 1. We have added a line in the paper to make this explicit (in Algorithm 1: Update $\phi$, $g$, $T$, and $f$ by minimizing the losses in Equations (5) and (6)).
> >
> > 4. The property of unique state embeddings is not theoretically guaranteed by the specification of our model. It is, however, required for Lemma 2. We, therefore, test this empirically, by verifying that no two states share the exact same embedded representation via all the experiments in Section 5. Since the state embeddings are initialized randomly and then updated in the training process, it is fairly unlikely that two states will ever share exactly the same embeddings. Nevertheless, the similarity between two embedded lower-dimensional representations is still captured by the l-1 or l-2 norm of their vector subtractions. We have added some more details on how we test this empirically in the paper to make this more clear (see Section 4.3.2. in the revised paper).
> >
> > 5. In our evaluation, we consider the gridworld and the Slotmachine as proof-of-concept experiments. Since these are fairly simple and for ease of exposition, we chose to only evaluate them using the comparatively simple VPG algorithm. For the more complex environments, i.e., the recommender system and the newly added Ant-v2, we conduct a broader range of experiments to show the robustness of our approach in combination with a range of RL algorithms.
> >
> > ## Additional feedback
> > Thanks for your feedback! We have incorporated these feedbacks in the revised paper, including the new title “Jointly-Learned State-Action Embeddings for Efficient Reinforcement Learning” as suggested to avoid any misunderstandings.

---

> ### Comment · Area_Chair1 · 2020-11-24
> **Please engage in the conversation**
>
> Dear reviewer,
>
> Please let us know whether the authors' rebuttal to your questions are satisfactory or not. If you need more clarifications on any issue, please ask your questions as soon as possible. Today (Nov. 24) is the last day that the authors can reply back to you.
>
> Thank you,
> AC

---

### Official Review · AnonReviewer2 · 2020-11-07
**Interesting work**

**Rating:** 6
**Confidence:** 2

**Review:**

Learning on environments with large state-action spaces can be difficult. This paper addresses this issue by learning a joint state-action embedding and learn an internal policy(\pi_i) on this embedded state-action space instead of the original state-action space. There are three parts of learning, 1. learning the embedding model that learns mapping from state to state embedding, 2. learning the internal policy, and 3. learning the mapping from action embedding to action space. The authors justify this approach by showing that the overall policy (\pi_o) can be expressed in terms of the internal policy (\pi_i). Furthermore, there is equivalence between the internal state-action-value function and overall state-action-value function and the authors show that updating \pi_i is equivalent to updating \pi_o.

The benefit of learning a policy on these joint state-action embedded space is that any policy gradient algorithm for continuous control can be used regardless of whether the original state-action is discrete or continuous. The authors claim that learning on this joint state-action embedded space is especially helpful in large discrete state-action spaces because relationships between state and action are often not clear with discrete representations. Their algorithm is compared on several environments highlighting this benefit, and also on several other benchmark domains.

Overall, the paper is well-written with an interesting approach to tackle the learning problem in large state-action space. They propose learning an internal policy backed by theoretical proof that show this is equivalent. Their results especially on the gridworld domain and slotmachine showcase the purported benefits of using their joint state-action space embedding (JSAE). I would overall recommend a weak accept, and I think strengthening the experiment part would make this a much better paper.

First, most of the experiments focus on VPG compared to PPO and SAC. While this is okay, there isn’t any citation on VPG or explanation of its exact implementation to help the readers follow what was actually used. Furthermore I think a justification on using VPG would be very helpful. Was it because it is a simple algorithm without complex structure? No replay buffer is necessary?

Second, I think there could be more discussion and insights into the various hyperparameters for JSAE. Figure 2a and 2c really highlights the benefit of using JSAE for large discrete state-action space domains. The authors hypothesize that it’s because the joint embedded space helps learn the relationship between states and actions. But were there any effects due to reduced embedded space? Would having a larger embedded space help capture the state-action relationship better? The experiment fixes the state and action embedding dimensions to a small value for the grid environments, and also only sweep a small range for other environments. A discussion on how these sweep ranges were chosen (e.g. was having small embedded space always helpful?), and other observations that the authors found while running the experiments would improve the paper a lot.

For other comments (won’t affect my decision but only for suggestions for improvement), I don’t think the experiment on HalfCheetah comparing VPG and VPG-JSAE shows much other than the fact that JSAE does not harm performance. SAC is known to have great performance on these continuous mujoco domains and it would be better to use SAC as the baseline and compare SAC-JSAE. HalfCheetah is also known to be one of the easy environments in Mujoco; I would be curious to know whether SAC-JSAE shows large improvements in high dimensional domains like Humanoid. Perhaps JSAE may capture a better state-action relationship in its embedded space and learn faster.

I think also reporting the best hyperparameters found for each agent and experiment would be helpful as well.

---

> ### Author Response · Authors · 2020-11-24
> **Response: Feedback and Questions**
>
> Thank you for your comments!
>
> **Regarding the use of VPG as a baseline:** We use this for our proof-of-concept due to the simplicity of VPG, i.e., few parameters to optimize. Our implementation is based on the reference implementation from OpenAI Spinninup (Achim, 2018). We have added this reference for clarity in the paper (see the revised paper).
>
> **Regarding the range sweeped for the embedding dimensionality:** We chose to keep the dimensionality of the embedding spaces fixed for the proof-of-concept gridworld environment as this is a simple environment that we mainly use to illustrate the functionality of our approach. For the other environments, we keep the dimensionality of the embedding spaces smaller than that of the original state and action spaces and try to cover a reasonable range of low values for this parameter. In choosing the range to sweep for this parameter, we consider the complexity of the environment, i.e., we expect more complex domains to require a higher dimensional embedding space. However, this is a hyperparameter that might require careful tuning on other problem domains. To illustrate the learned embeddings, we have also added a visualization of these for the gridworld domain in the paper (see Fig. 2 in the revised paper).
>
> **Experiment on more challenging environments:** In addition to our experiments on the HalfCheetah environment, we have now conducted some **new experiments on the more complex environment Ant-v2**, using SAC, PPO, and VPG as benchmarks. The results are included in the paper (Fig. 3 in the revised paper) and demonstrate that our approach outperforms the benchmarks. In general, we find that the JSAE methodology we propose works particularly well for large discrete state-action spaces. In continuous domains, i.e., the continuous-state gridworld, half-cheetah, and Ant-v2 environments, we observe that our approach works well for more complex domains, specifically for the Ant-v2 environment. Since continuous domains tend to have some inherent structure, e.g. force applied to joints in the HalfCheetah or Ant environment has some inherent order, it tends to be more difficult for our embedding model to capture structure beyond what is already present in the continuous state and action representations. However, for more complex domains (Ant-v2), the RL agent further benefits from the reduced complexity that results from our embedding approach (lower dimensionality of state and action representation and the structure of the embedding spaces that captures the similarity between different states and actions), and thus converges more quickly and also achieves better performance at the end.
> We have added the best hyperparameters found for each of the experiments to the appendix. We hope that this facilitates easier reproducibility. We also plan to publish the code after some polishing.
>
> (References are in the paper)

---

> > ### Comment · AnonReviewer2 · 2020-11-24
> > **Paper not updated**
> >
> > Hi, I don't think the paper has been updated.
> > I cannot find results for Ant-v2 in Fig. 3 or visualization of learned embeddings in Fig 2.

---

> > > ### Author Response · Authors · 2020-11-24
> > > **Paper Updated**
> > >
> > > The revised paper has now been uploaded. Apologies for the delay between uploading our responses to the reviews and uploading the revised version of the paper.

---

> > ### Comment · AnonReviewer2 · 2020-11-24
> > **Thanks for the update!**
> >
> > Thank you for doing more runs and adding Ant-v2 results.
> > JSAE seems to give large improvements for domains with much larger state dimension despite being a continuous state/action space where there is already association to similar states/actions.
> >
> > After a careful look however, HalfCheetah-v2 and Ant-v2 experiments were conducted completely differently (mentioned in A.4.2).
> > HalfCheetah-v2 embeddings were pre-trained with 30,000 samples and fixed, while Ant-v2 embeddings were pre-trained with 100,000 samples and allowed updates during training.
> >
> > I think this difference is worth mentioning in the main paper, instead of in the appendix, because the readers may assume the improvement difference was only due to the complexity of the environment rather than due to the different training method.
> >
> > I now think experiments in HalfCheetah and Ant shows more about the benefit of pre-training with more samples and allowing continuous updates.
> >
> > Some minor comments:
> > I think the labels in Ant-v2: SAC Embed should be changed to SAC-JSAE.
> > For the final version I think it'll improve readability to have same methods in same line colors.

---

> > > ### Author Response · Authors · 2020-11-24
> > > **Response to Update**
> > >
> > > Thanks for your additional comments!
> > >
> > > We have incorporated your comments in the revised version of the paper.
> > > We now mention for which experiments continuous updates are enabled at the beginning of Section 5. Hopefully, that helps to put the results into context.
> > >
> > > Thank you for spotting the mistake in the Figure! We have adjusted the labels accordingly.

---

### Author Response · Authors · 2020-11-24
**Summarization of major changes in the revised paper**

1. **Positioning of our approach in relation to dominant branches of research in state representation learning:** In response to the comments we received, we have included a discussion of world models and state aggregation techniques using bisimulation metrics in Section 3 (Related Work). We also position our approach in relation to this and point out similarities and differences between our method and previous literature pertaining to the two mentioned areas.

2. **Additional experiment on a more complex continuous domain:** We have conducted an additional experiment on the Ant-v2 environment from the Mujoco physics engine. This domain is substantially more complex than the previously included half-cheetah environment. We find that our approach outperforms the baseline in this environment (see Section 5 (Empirical Evaluation)).

3. **Experiments ran with more random seeds:** Instead of evaluating the performance of our proposed approach and the chosen baselines using 5 random seeds, we now evaluate this using 10 random seeds (see Section 5 (Empirical Evaluation)). The conclusions that can be drawn from the empirical evaluation do not change as a consequence of this.

4. **Visualisation of learned embeddings:** We have included an illustrative visualization of the learned state and action embeddings for a discrete gridworld domain in the paper. This serves to illustrate that the structure of state and action spaces is captured well by our proposed embedding model (see Section 5 (Empirical Evaluation)).

5. **Inclusion of full hyperparameters:** We have added the best hyperparameters obtained from the grid searches for all of our experiments to the appendix to foster reproducibility.

6. **Clarification of Assumptions and some minor points:** We have added additional details on the empirical tests performed to verify Assumptions 1 and 3 in the paper (Section 4.2 and Section 4.3.2.).

---

### Decision · Program_Chairs · 2021-01-07
**Final Decision**

**Decision:**

Reject

**Comment:**

Most reviewers believe that the paper is not ready for publication. Among their concerns are:
- whether the new experiment with 10 runs are conducted correctly,
- the significance of the theoretical part,
- correctness of Lemma 2,
- generalization claims may not follow from the theoretical results,
- comparison with Zhang et al. (2020).

Given these and the lack of support from reviewers, unfortunately I cannot recommend acceptance of this paper at this stage. I encourage the authors to improve their paper according to these concerns.

I copy-paste some of the comments that came after Nov. 24th. The authors might want to use them to improve their work.
---
First, given the assumptions, the theorems/lemmas are not sufficient to be a solid contribution. I think what important is, can the Alg 1 lead to the optimal policy? More specifically, I do not doubt lemma 2; I am concerned about if updating the representations in an online manner (it should be highly nonstationary) can result in the optimal policy. Some two-time scale analysis may address this question. As an additional note, since f can be a many-to-one mapping, policy pi_i is a multimodal distribution. I am unsure if the authors consider this during implementation.

Second, without the two-time scale analysis, I would give weak acceptance if the authors can persuade that the superior performances are indeed due to the proposed jointly embedding learning method. That's why I ask for a baseline that directly considers environment model learning as an auxiliary task.

In the abstract, the authors state that "In this work, we propose a new approach for jointly learning embeddings for states and actions ...," in fact, this is not new. Almost any deep RL algorithm can be thought of as the process of learning state and/or action representations. In the response, the authors say, "We do not believe ..., as ... are embedded into the same space." How to do embedding is more like an implementation issue; one can encode them into different spaces and learn them by learning an environment model. It is nothing fancy/novel.

I consider this paper's main novelty to learn the optimal policy in the embedded state and action spaces, and the embeddings are learned by environment model learning. Thus, the authors need to have strong evidence to persuade people: 1) using an environment model to learn the embeddings is really useful; 2) a separate process of learning the policy in the embedded spaces is essential. Such evidence is necessary to make this paper a solid contribution.

Learning an environment model has been used as an auxiliary task in deep RL. Using such a baseline is to validate that it is necessary to learn the policy in the embedded space separately. The authors should also actively design other baselines to substantiate their claims.